# Time-variable gravity fields and ocean mass change from 37 months of kinematic Swarm orbits

Christina Lück[1], Jürgen Kusche[1], Roelof Rietbroek[1], and Anno Löcher[1]

[1]Institute of Geodesy and Geoinformation, University of Bonn, Bonn, Germany

*Correspondence to:* lueck@geod.uni-bonn.de

**Abstract.** Measuring the spatiotemporal variation of ocean mass allows for partitioning volumetric sea level change, sampled by radar altimeters, into mass-driven and steric parts. The latter is related to ocean heat change and the current Earth's energy imbalance. Since 2002, the Gravity Recovery and Climate Experiment (GRACE) mission provides monthly snapshots of the Earth's time-variable gravity field, from which one can derive ocean mass variability. However, GRACE has reached the end of its lifetime with data degradation and several gaps occurred during the last years, and there will be a prolonged gap until the launch of the follow-on mission GRACE-FO. Therefore, efforts focus on generating a long and consistent ocean mass time series by analyzing kinematic orbits from other low-flying satellites, i.e. extending the GRACE time series.

Here we utilize data from the European Space Agency's (ESA) Swarm Earth Explorer satellites to derive and investigate ocean mass variations. For this aim, we use the integral equation approach with short arcs (Mayer-Gürr, 2006) to compute more than 500 time-variable gravity fields with different parameterizations from kinematic orbits. We investigate the potential to bridge the gap between the GRACE and the GRACE-FO mission and to substitute missing monthly solutions with Swarm results of significantly lower resolution. Our monthly Swarm solutions have a root mean square error (RMSE) of $4.0\,\mathrm{mm}$ with respect to GRACE, whereas directly estimating constant, trend, annual and semiannual (CTAS) signal terms leads to an RMSE of only $1.7\,\mathrm{mm}$. Concerning monthly gaps, our CTAS Swarm solution appears better than interpolating existing GRACE data in 13.5 % of all cases, when artificially removing one solution. In case of an 18-months artificial gap, 80.0 % of all CTAS Swarm solutions were found closer to the observed GRACE data compared to interpolated GRACE data. Furthermore, we show that precise modelling of non-gravitational forces acting on the Swarm satellites is the key for reaching these accuracies. Our results have implications for sea level budget studies, but they may also guide further research in gravity field analysis schemes, including satellites not dedicated to gravity field studies.

## 1 Introduction

Sea level rise, currently about $3\,\mathrm{mm}\,\mathrm{yr}^{-1}$ in global average, will affect many countries and communities along the world's coastlines, with potentially devastating consequences (Nicholls and Cazenave, 2010; Stocker et al., 2013). Knowing ocean mass change is important because it enables the partitioning of volumetric sea level changes, as measured by radar altimeters, into mass and steric parts. The steric sea level change is related to ocean heat content and thus to the question whether the

Earth's energy imbalance (currently 0.9 W m$^{-2}$, Trenberth et al. 2014) can be explained. Yet, a number of studies found differing ocean mass rates from the GRACE data sets (Rietbroek et al., 2016; Cazenave and Llovel, 2010; Lombard et al., 2007; Gregory et al., 2013; Llovel et al., 2014). Therefore, and considering the gap between the GRACE and the GRACE-FO mission, alternative methods to derive ocean mass changes are expected to provide valuable insights. As we will see in the course of this paper, the ESA Swarm Earth Explorer mission (Friis-Christensen et al., 2008) is able to detect regular as well as non-regular ocean mass changes such as La Niña events.

Swarm was successfully launched into a near-polar low Earth orbit (LEO) on November 22, 2013. The three identical satellites, referred to as Swarm A, Swarm B and Swarm C, were designed to provide the best-ever survey of the geomagnetic field and its temporal variability. The attitude of each satellite is measured by star trackers with three camera head units. For Precise Orbit Determination (POD), each spacecraft is equipped with an 8-channel dual-frequency GPS receiver (Zangerl et al., 2014) and laser retroreflectors that allow satellite laser ranging (SLR) for orbit validation. Also, all three satellites carry accelerometers for deriving the non-gravitational accelerations, which would have been helpful in gravity field determination. However, these measurements were found to exhibit spurious signals, mostly thermal related, and cannot be used in a straightforward way. Siemes et al. (2016), after reprocessing, provide corrected non-gravitational accelerations in along-track direction for Swarm C, but it is unclear whether such corrections will be ever derived for all components.

Swarm A and C fly side by side in a mean altitude of 450 km while the Swarm B orbit is at 515 km at present; this results in a drifting of Swarm B's orbital plane with respect to the orbital planes of the other two satellites. This constellation, together with the global coverage due to near-polar and near-circular orbits, provides the opportunity for gravity field recovery. This has sparked a renewed interest in satellite gravity method development, in particular since the GRACE mission has reached the end of its lifetime and its follow-on GRACE-FO will be launched only in spring 2018. At the time of writing, kinematic LEO orbits are considered as a promising option for deriving global gravity fields during a GRACE mission gap (Gunter et al., 2009; Weigelt et al., 2013; Rietbroek et al., 2014). Several Swarm simulation studies had already been conducted before launch, e.g. Gerlach and Visser (2006) and Visser (2006). Wang et al. (2012), using the energy integral approach, suggested that static gravity solutions could be derived up to degree 70 from Swarm-like constellations, while time-variable monthly fields might be recovered up to degree 5 to 10. These authors furthermore hypothesized that the use of kinematic baselines would increase the spatial resolution, albeit at the expense of weaker solutions at longer wavelengths. However, Jäggi et al. (2009) showed with real GRACE GPS-derived baselines that the benefit will probably be small. Consequently, after launch, kinematic GPS orbits have been derived and used by different groups to estimate time-variable gravity fields: Teixeira da Encarnação et al. (2016) compare solutions of the Astronomical Institute of the University of Bern (AIUB, Jäggi et al. 2016), the Astronomical Institute of the Czech Academy of Science (ASU, Bezděk et al. 2016) and the Institute of Geodesy (IFG) of the Graz University of Technology (Zehentner, 2017), suggesting that a meaningful monthly time-varying gravity signal can be derived until degree 12, considering the average of the three models.

| | product | sampling | availability | reference frame |
|---|---|---|---|---|
| Kinematic Orbits | ESA Level 2 KIN (van den IJssel) | 10s | December 1st, 2013 to July (A:15th., B:15th., C:10th.), 2014 | ITRF 2008 |
| Kinematic Orbits | ESA Level 2 KIN (van den IJssel) | 1s (10s is used) | July (A:15th., B:15th., C:10th.), 2014 to December 31st, 2016 | ITRF 2008 |
| Star Camera | ESA Level 1b | 1s (10s is used) | December 1st, 2013 to December 31st, 2016 | ITRF 2008 to satellite frame |

**Table 1.** Utilized orbit and star camera data.

In this study, we first compute a set of in-house time-variable gravity fields from Swarm kinematic orbits to further derive a time series of ocean mass change. To this end, we use the integral equation approach developed earlier at the University of Bonn (Mayer-Gürr, 2006) and compare time series of monthly Swarm gravity solutions and CTAS solutions to existing GRACE solutions. We model non-gravitational accelerations (drag, solar radiation pressure and Earth radiation pressure) for all three Swarm satellites. This has been found to be important to improve the gravity field results.

This article is organized as follows: In Sect. 2 we describe the used data sets and background models, followed by a brief discussion of methods in Sect. 3. Section 4 will present our results for ocean mass change, discuss the effects of non-gravitational force modelling and gravity field parameterization, and the relative contribution of the three satellites.

## 2 Data

### 2.1 Swarm data

Time series of quality-screened, calibrated and corrected measurements are provided in the Swarm Level 1b products. The Swarm Satellite Constellation Application and Research Facility (SCARF, Olsen et al. 2013) further processes Level 1b data and auxiliary data to Level 2 products. Here we use Level 2 kinematic orbits (van den IJssel et al., 2015, 2016), see Table 1, and Level 1b star camera data which are required for transforming from the terrestrial to satellite reference frame. During the processing, the satellite reference frame needs to be referred to the inertial frame; this is achieved by multiplying the rotation matrix derived from the star camera data with the Earth rotation matrix (Petit and Luzum, 2010).

For modelling non-conservative forces, we implemented a Swarm macro model consisting of area, orientation and surface material for 15 panels, supplemented with surface properties such as diffuse and specular reflectivity (ESA, Christian Siemes, personal communication) for computing solar radiation pressure and Earth radiation pressure consisting of measured albedo and emission.

| Background model | Product | Reference |
|---|---|---|
| Static Field | GOCO05c | Pail et al. (2016) |
| Earth Rotation | IERS2010 | Petit and Luzum (2010) |
| Moon, Sun and Planets | JPL DE421 | Folkner et al. (2009) |
| Earth Tide | IERS2010 | Petit and Luzum (2010) |
| Ocean Tide | EOT11a | Savcenko and Bosch (2012) |
| Pole Tide | IERS2010 | Petit and Luzum (2010) |
| Ocean Pole Tide | Desai2004 | Petit and Luzum (2010) |
| Atmospheric Tides | van Dam / Ray | van Dam and Ray (Updated October 2010) |
| Atmosphere and Ocean Dealiasing | AOD1B RL05 | Flechtner et al. (2015) |
| Permanent Tidal Deformation | included (zero tide) | |

**Table 2.** Background models used during the processing

## 2.2 Background models

During gravity field recovery, we used the GOCO05c model (Pail et al., 2016) up to degree 360 as a mean background field. All time-variable background models (cf. Table 2) are consistent with GRACE RL05 processing standards (Dahle et al., 2012) except for the atmospheric tides which were chosen such as to be aligned with the Graz ITSG-Grace2016 solutions. The reason
for this is that we compare our Swarm solutions to the monthly ITSG-Grace2016 solutions (Mayer-Gürr et al., 2016).

## 2.3 Density model

Drag modelling requires knowing the thermospheric density and temperature. In this work, we make use of the empirical NRLMSISE-00 model (Picone et al., 2002). NRLMSISE-00's data base includes total mass density from satellite accelerome-
ters and POD, temperature from incoherent scatter radar as well as molecular oxygen number density, however collected under different solar activity conditions. The model is driven by observed solar flux (F10.7 index) and geomagnetic index ($A_P$). In Vielberg et al. (subm.) we compare NRLMSISE-00 to GRACE-derived thermospheric density and derive an empirical correction for this model; this has not yet been applied here.

## 3 Methods

In order to address our central question - to what extent will Swarm enable one to infer ocean mass change - we first compute time-variable gravity fields from kinematic orbits, while considering different processing options. Then, ocean mass is derived from the computed Stokes coefficients (e.g. Chambers and Bonin, 2012), and results will be compared to the ITSG-GRACE solutions.

In the following, we describe our modelling of the non-conservative forces (Sect. 3.1), the processing method (integral equation approach with short arcs), and two options for gravity field parameterization within the gravity recovery: (1) estimation of monthly fields (2) estimation of a CTAS model for each harmonic coefficient from the whole mission lifetime in a single adjustment (Sect. 3.2). Finally, results are compared to the ITSG-GRACE solution in terms of area averages for the total ocean as well as - for comparison - to water storage change within various large terrestrial river basins (Sect. 3.3).

## 3.1 Modelling of non-gravitational forces

While all three Swarm satellites carry accelerometers intended to support POD and the study of the thermosphere, unfortunately these data have turned out as severely affected by sudden bias changes ("steps") and temperature-induced bias variations. Siemes et al. (2016) developed a method to clean and calibrate the along-track acceleration of Swarm C. However, Swarm A and B (the former to a lesser extent than the latter), as well as the other C directions are affected by serious issues and it is not clear whether these data can be used in gravity field applications. In the light of recent improvements of empirical thermosphere models (Vielberg et al., subm.) and seen that we require all three components of non-conservative acceleration $\mathbf{a}_{model}$ for gravity recovery, we decided to rather model them, using the well-known relation

$$\mathbf{a}_{model} = \mathbf{a}_{drag} + \mathbf{a}_{SRP} + \mathbf{a}_{ERP}. \tag{1}$$

$\mathbf{a}_{model}$ is the sum of atmospheric drag $\mathbf{a}_{drag}$, solar radiation pressure $\mathbf{a}_{SRP}$ and Earth radiation pressure $\mathbf{a}_{ERP}$. We will briefly summarize our implementation below.

*Atmospheric Drag*:
Atmospheric drag is commonly taken into account by evaluating

$$\mathbf{a}_{drag} = C_d \frac{A_{ref}}{2m} \rho v_r^2 \hat{\mathbf{v}}_r, \tag{2}$$

where $m$ is its mass, $\rho$ the thermospheric density (here from NRLMSISE-00), $v_r$ the velocity of the satellite relative to the atmosphere, and $\hat{\mathbf{v}}_r$ the normalized velocity vector relative to the atmosphere. $A_{ref}$ is a reference area that cancels out in the computation of $C_d$ (more precisely in the computation of $C_{D,i,j}$ and $C_{L,i,j}$, which will be introduced later), where the ratio of the area of each plate to $A_{ref}$ is taken into account. $C_d$ is evaluated as the sum over each plate $i$ and each constituent of the atmosphere $j$

$$C_d = \left[ \sum_{i=1}^{N} \sum_{j=1}^{M} \frac{\rho_j}{\rho} \left( C_{D,i,j} \hat{\mathbf{u}}_D + C_{L,i,j} \hat{\mathbf{u}}_{L,i} \right) \right] \cdot \hat{\mathbf{v}}_r, \tag{3}$$

where the contributions of drag $C_{D,i,j}$ and lift $C_{L,i,j}$ are evaluated separately with their associated unit vectors $\hat{\mathbf{u}}_D$ and $\hat{\mathbf{u}}_{L,i}$. We follow Sentman et al. (1961) for further computations of $C_{D,i,j}$ and $C_{L,i,j}$.

*Solar Radiation Pressure*:

Solar radiation is absorbed or reflected at the satellite's surface, leading to an acceleration (Sutton, 2008; Montenbruck and Gill, 2005)

$$\mathbf{a}_{SRP} = \sum_{i=1}^{N} -\nu \frac{1AU^2 R A_i \cos(\Phi_{inc,i})}{r_{\odot}^2 m} \left[ 2\left(\frac{C_{rd,i}}{3} + c_{rs,i}\cos(\Phi_{inc,i})\right)\hat{\mathbf{n}}_i + (1 - c_{rs,i})\hat{\mathbf{s}} \right]. \tag{4}$$

5   Equation 4 accounts for SRP over each of the $N$ plates of the macro model. $R$ is the solar flux constant valid at distance of 1 astronomical unit (AU), $A_i$ is the area of the $i$th plate and $c_{rd,i}$ and $c_{rs,i}$ are the diffuse and specular reflectivity coefficients. $\Phi_{inc,i}$ denotes the angle between the the Sun (unit vector $\hat{\mathbf{s}}$) and the normal vector of each panel $\hat{\mathbf{n}}_i$. The shadow function $\nu$ varies between 0 when the satellite is in eclipse and 1 if it is fully illuminated. The term $1AU^2/r_{\odot}^2$ accounts for the eccentricity of the Earth's orbit, with $r_{\odot}$ being the varying Sun-satellite distance.

*Earth Radiation Pressure*:

Radiation emitted from the Earth's surface (ERP) is taken into account similar to solar radiation pressure:

$$\mathbf{a}_{ERP} = \sum_{i=1}^{N}\sum_{j=1}^{M} -\frac{R^j A_i \cos\left(\Phi_{inc,i}^j\right)}{m} \left[ 2\left(\frac{C_{rd,i}}{3} + c_{rs,i}\cos\left(\Phi_{inc,i}^j\right)\right)\hat{\mathbf{n}}_i + (1 - c_{rs,i})\hat{\mathbf{s}}^j \right]. \tag{5}$$

The satellite's footprint is divided into $M$ sections and $R^j$ takes into account the effect of albedo and emission (we use the
Cloud and the Earth's Radiant Energy System (CERES) dataset EBAF-TOA Ed2.8 that provides monthly values (Loeb et al., 2009)). Different from the conventional implementation (Knocke et al., 1988), we expanded these data into a low-degree spherical harmonic representation to account for longitudinal variations.

## 3.2   Gravity field estimation

For gravity field estimation, we use the integral equation approach (Schneider, 1968; Reigber, 1969). Kinematic orbits are
partitioned into (short) arcs and each 3D position $\mathbf{r}(\tau)$ between the arc's begin and end ($\mathbf{r}_A$ and $\mathbf{r}_B$) can be expressed as

$$\mathbf{r}(\tau) = \mathbf{r}_A(1-\tau) + \mathbf{r}_B\tau - T^2 \int_0^1 K(\tau,\tau')\mathbf{f}(\tau')d\tau', \tag{6}$$

with normalized time $\tau$ and the integral kernel

$$K(\tau,\tau') = \begin{cases} \tau'(1-\tau) & \text{for } \tau' \leq \tau \\ \tau(1-\tau') & \text{for } \tau' > \tau. \end{cases} \tag{7}$$

In other words, $T^2 \int_0^1 K(\tau,\tau')\mathbf{f}(\tau')d\tau'$ in Eq. 6 represents the offset of the current position from a straight line connecting $\mathbf{r}_A$
and $\mathbf{r}_B$, caused by gravitational and non-gravitational forces $\mathbf{f}(\tau')$. After discretization (sampling rate of kinematic orbits is 1

second after July 2014), one can write the above as an adjustment problem with two groups of solved-for parameters:

$$
\mathbf{y} = \begin{bmatrix} \mathbf{r}_A \\ \mathbf{r}_B \\ \mathbf{acc}_{perArc} \end{bmatrix} \text{ and } \mathbf{x} = \begin{bmatrix} c_{20} \\ c_{21} \\ s_{21} \\ \vdots \\ s_{nn} \\ \mathbf{acc}_{global} \end{bmatrix}. \tag{8}
$$

$\mathbf{y}$ contains all arc-related parameters, which can be eliminated from the normal equation system during the estimation. These include start and end position of each arc and additional parameters such as accelerometer bias or scale factors. The gravity field parameters are then collected in $\mathbf{x}$. For more details of the integral equation approach, see Mayer-Gürr (2006) and Löcher (2010).

In this study, we consider two different ways of parameterizing the gravity field: (1) To be consistent with GRACE, we estimate monthly spherical harmonic coefficients complete to varying low degrees. (2) CTAS solution: As we aim at a long and stable time series, we additionally parameterize a set of trends and (semi)annual harmonic amplitudes to the constant part for each Stokes coefficient in a single adjustment:

$$
\begin{aligned}
c_{nm}(t) &= \bar{c}_{nm} + \dot{c}_{nm}(t - t_0) \\
&\quad + c_{nm}^{c_1} \cos\left( 2\pi \frac{t - t_0}{yr} \right) + c_{nm}^{s_1} \sin\left( 2\pi \frac{t - t_0}{yr} \right) \\
&\quad + c_{nm}^{c_2} \cos\left( 4\pi \frac{t - t_0}{yr} \right) + c_{nm}^{s_2} \sin\left( 4\pi \frac{t - t_0}{yr} \right), \\
s_{nm}(t) &= \bar{s}_{nm} + \dot{s}_{nm}(t - t_0) \\
&\quad + s_{nm}^{c_1} \cos\left( 2\pi \frac{t - t_0}{yr} \right) + s_{nm}^{s_1} \sin\left( 2\pi \frac{t - t_0}{yr} \right) \\
&\quad + s_{nm}^{c_2} \cos\left( 4\pi \frac{t - t_0}{yr} \right) + s_{nm}^{s_2} \sin\left( 4\pi \frac{t - t_0}{yr} \right).
\end{aligned} \tag{9}
$$

We estimate the spherical harmonic coefficients from degree 2 onward. As described in Sect. 3.1 we derive non-gravitational accelerations from models, which we then use in the gravity field estimation as a proxy for accelerometer measurements. Due to the presence of errors, e.g. caused by uncertainties in the density model or errors in the macro model, the resulting non-gravitational accelerations might not always reflect the truth. To prevent these errors from propagating into the gravity field estimates, it is common to introduce additional parameters. Here we co-estimate an "accelerometer bias" per arc and per axis, either as a constant value or with an additional trend parameter. While we found this usually sufficient, we also performed tests with an additional global scaling factor per axis. Another possibility that is also evaluated in this paper is to co-estimate the bias globally. The influence of this "accelerometer parameterization" will be evaluated in the course of this paper, yet one needs to bear in mind that these parameters rather measure force model inconsistencies and should not be mixed up with instrument

errors. We furthermore investigate the influence of different arc lengths (which affects the temporal acceleration parameterization) as well as the effect of spherical harmonic truncation.

## 3.3 Ocean mass changes and river basin averages

5  As was mentioned already, we choose different regions for our investigation (see Fig. 1, but our focus is on the total ocean in order to test the hypothesis that Swarm can bridge the GRACE ocean mass time series.

For computing smoothed basin mass averages, let $F(\lambda, \Phi)$ be the equivalent water height (EWH), derived from the spherical harmonics (Wahr et al., 1998). The smoothed region average $\overline{F}_{O_W}$, considering the smoothing Kernel $W$ (here a 500 km Gaussian filter) over the region $O$ can be expressed as

$$\overline{F}_{O_W} = \frac{1}{\overline{O}_W} \int\limits_{\Omega} O_W(\lambda, \theta) F \, dw. \tag{10}$$

The integral is effectively evaluated for the smoothed area function $O_W(\lambda, \theta)$,

$$O_W(\lambda, \theta) = \sum_{n=0}^{\infty} \sum_{m=-n}^{n} \overline{O}_{nm}^{W} \overline{Y}_{nm}(\lambda, \theta) = \frac{1}{4\pi} \int\limits_{\Omega} W(\lambda, \theta, \lambda', \theta') O(\lambda', \theta') \, dw'. \tag{11}$$

Some postprocessing needs to be applied to the estimated gravity fields, depending on the application. As we compare our results to the monthly GRACE solutions, we test replacing the $c_{20}$ coefficient with those derived from satellite laser ranging

(SLR) (Cheng et al., 2013). While replacing $c_{20}$ leads to a workflow more in line with GRACE, keeping the Swarm-derived $c_{20}$ would answer the question whether Swarm alone is able to measure mass change relative to a reference (here GOCO05c). In a next step, we add all degree 1 coefficients to correct for geocenter motion (Swenson et al., 2008), which cannot be detected with the current GRACE and Swarm processing. We apply a correction for glacial isostatic adjustement following A et al. (2013), but as long as we apply the same correction to GRACE, the comparison between Swarm and GRACE will be independent of

this choice. We employed an ocean mask that includes the Arctic ocean and does not have a coastal buffer zone.

## 4  Results

If not stated differently, we used the parameterization in Table 3 for monthly ocean mass or ocean mass from a direct estimation of CTAS signal terms. We chose these parameterizations because they represent our best monthly solution (as will be seen in Fig. 11) and the best CTAS solution up to degree and order (d/o) 12 (see Fig. 10). The choice of the same degree allows a

comparison of the results. Our test studies include all possible combinations of the parameterizations shown in Table 4, which leads to more than 500 configurations.

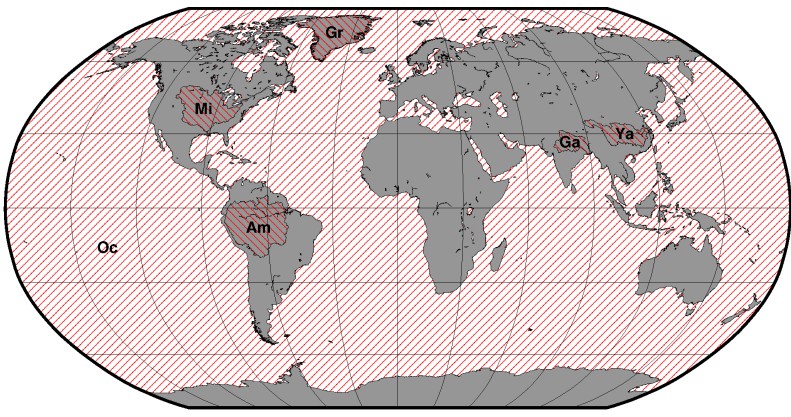

**Figure 1.** Areas of investigation: ocean (OC), Amazon (AM), Mississippi (MI), Greenland (GR), Yangtze (YA) and Ganges (GA). The boundaries are taken from the Food and Agriculture Organization of the United Nations (FAO)

| | arc length [minutes] | non-grav. acc. | bias | scale | maximum d/o |
|---|---|---|---|---|---|
| monthly | 30 | modelled | constant per arc (pA0) | none | estimated until 40, evaluated until 12 |
| CTAS | 45 | modelled | constant+trend per arc (pA1) | none | static until 40 (evaluated until 12), time-variable until 12 |

**Table 3.** Parameterization for our monthly solutions and for our estimation of CTAS signal terms. All results are subject to a 500 km Gaussian filter.

| arc length [minutes] | non-grav. acc. | bias | scale | | maximum d/o |
|---|---|---|---|---|---|
| 30 | not modelled | none | none | monthly: | estimated until 20/40, evaluated until 10/12/14 |
| 45 | modelled | constant per arc (perArc0) | global | CTAS: | static part until 20/40/60 (evaluated until 10/12/14), |
| 60 | | constant+trend per arc (perArc1) | | | time-variable part until 10/12/14 |
| | | constant global (global0) | | | |
| | | polyn. of deg. 4 global (global4) | | | |

**Table 4.** Parameterizations that have been tested in this study. This table should not be read row-wise. It lists all possible choices for each heading. One solution can consist of any combination of the entries, for example: a monthly solution with an arc length of 60 minutes, modelled non-gravitational accelerations, a constant global bias, no scaling factor, max. d/o estimated: 40, evaluated until d/o 10.

## 4.1 Ocean mass from GRACE and Swarm

Figure 2 shows monthly ocean mass change in mm EWH derived from GRACE as a reference and from different Swarm time-variable gravity (TVG) solutions from AIUB, ASU, IfG and the Institute of Geodesy and Geoinformation (IGG) in Bonn (processing details can be found in Table 5). The IGG time-variable gravity field was computed with an arc length of 30 minutes
and with modelled non-gravitational accelerations, with a constant bias per arc and direction being co-estimated, which leads to our best solution. All Swarm time series show a behavior similar to the GRACE solution, but they appear overall noisier, as can be seen from the variances in Table 6. The quality of all solutions improves after the GNSS receiver update in July 2014. The impact of tracking loop updates on gravity field recovery is discussed in Dahle et al. (2017). It is furthermore interesting to compute the RMSE of all solutions when we assume the GRACE solution to be the truth (first row of Table 6). The ASU
time series has the lowest RMSE of 2.8 mm; it is closest to GRACE. The IGG solution has the second lowest RMSE of 4.0 mm. To assess the spread between the different Swarm solutions, we compute the RMSE for each combination (off-diagonal of Table 6) which is of the same magnitude as the RMSEs of GRACE and Swarm.

An important issue in extending the ocean mass time series is the accuracy of the trend. Table 7 shows the trends as well as
the amplitude and phase of the Swarm solutions. The trend of the IGG solution ($3.3$ mm yr$^{-1}$) is the closest to GRACE ($3.5$ mm yr$^{-1}$). While the trend over three years itself cannot be considered as representative for the GRACE era due to interannual variability of barotropic modes, this suggests that Swarm data could be used to bridge a gap between GRACE and GRACE-FO.

Figure 3 shows the degree variances and the difference degree variances of GRACE and our IGG solution for May 2016 with
20 respect to our reference field GOCO05c. Obviously, the higher the degree, the higher is the discrepancy between GRACE and Swarm. The difference (dotted gray line) indicates that for this particular month Swarm is only reliable for degrees up to about 10, which is due to the much lower precision of the GPS data compared to the GRACE inter-satellite K-Band ranging. Since the formal errors (dotted black line) are not calibrated, they are too optimistic and always lower than the difference between GRACE and Swarm. This will be addressed in the future by including realistic covariance information of the kinematic orbits.
As Fig. 3 only shows the degree variances for one particular month, we investigate different maximum degrees in the following (see Table 4). We evaluate our monthly fields until d/o 10, 12 or 14. Even though higher degrees do not contribute a reasonable time-variable signal, we estimate the monthly fields until d/o 20 or 40, because high degrees can absorb errors that would otherwise propagate in the lower degrees. For our CTAS solution, we estimate a static part ($\bar{c}_{nm}$ and $\bar{s}_{nm}$ in Eq. 9) until d/o 20, 40 or 60, while the time-variable part is estimated until d/o 10, 12 or 14.

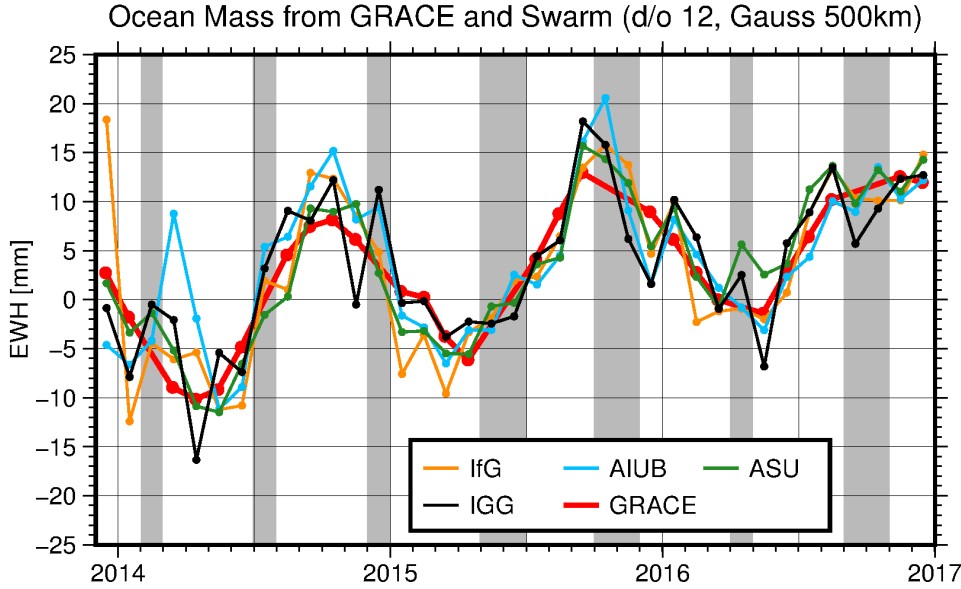

**Figure 2.** Ocean Mass from ITSG-Grace2016 and Swarm. GRACE data gaps are highlighted in gray.

|         | AIUB              | ASU          | IGG        | IfG        |
|---------|-------------------|--------------|------------|------------|
| Orbit   | AIUB              | ITSG         | ESA        | IfG        |
|         | (screened version)|              |            |            |
| Approach| Celestial         | Acceleration | Short-Arc  | Short-arc  |
|         | mechanics         | approach     | approach   | approach   |
|         | approach          |              |            |            |
| max d/o | 70                | 40           | 40         | 60         |

**Table 5.** Comparison of Swarm solutions from different institutes: Orbit product, computing method, and maximum degree and order provided.

|       | GRACE | AIUB | ASU | IGG | IfG |
|-------|-------|------|-----|-----|-----|
| GRACE | 6.6   | 5.1  | 2.8 | 4.0 | 5.2 |
| AIUB  |       | 7.4  | 4.5 | 4.3 | 5.4 |
| ASU   |       |      | 7.3 | 4.2 | 4.1 |
| IGG   |       |      |     | 7.5 | 5.6 |
| IfG   |       |      |     |     | 8.5 |

**Table 6.** Comparison of the variance [mm] of the individual ocean mass time series (main diagonal) and the RMSE [mm] between two solutions (off-diagonal). The results are based on the time series of Fig. 2.

|                                | GRACE | AIUB         | ASU          | IGG          | IfG          |
|--------------------------------|-------|--------------|--------------|--------------|--------------|
| Trend [mm yr$^{-1}$]           | 3.5   | 2.1 (3.2)    | 4.2 (4.6)    | 3.3 (4.3)    | 2.4 (3.2)    |
| Amplitude (annual) [mm]        | 7.9   | 7.4 (7.6)    | 6.9 (8.1)    | 6.8 (7.8)    | 9.0 (10.1)   |
| Amplitude (semiannual) [mm]    | 1.1   | 2.9 (4.5)    | 0.5 (0.8)    | 2.3 (0.8)    | 1.2 (1.9)    |
| Phase (annual) [days]          | -12.0 | -12.4 (-11.7)| -12.1 (-12.4)| -12.8 (-12.4)| -12.6 (-12.2)|
| Phase (semiannual) [days]      | 6.6   | 13.4 (13.2)  | 13.7 (12.4)  | 7.8 (12.4)   | -9.9 (-12.2) |

**Table 7.** Comparison of Swarm solutions from different institutes: trend, amplitude and phase. The values in brackets indicate the results for the exact same months that are available for GRACE, while the values without brackets are computed from the whole Swarm time series. The results are based on the time series of Fig. 2.

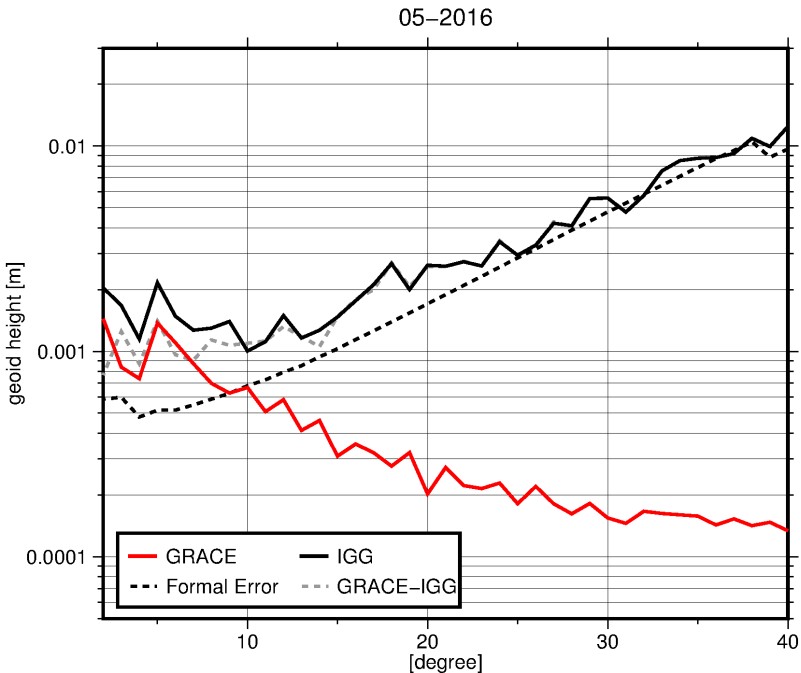

**Figure 3.** Degree variances for GRACE and Swarm (solution May 2016). Formal errors as well as the difference degree variance (GRACE-Swarm) are shown with dotted lines.

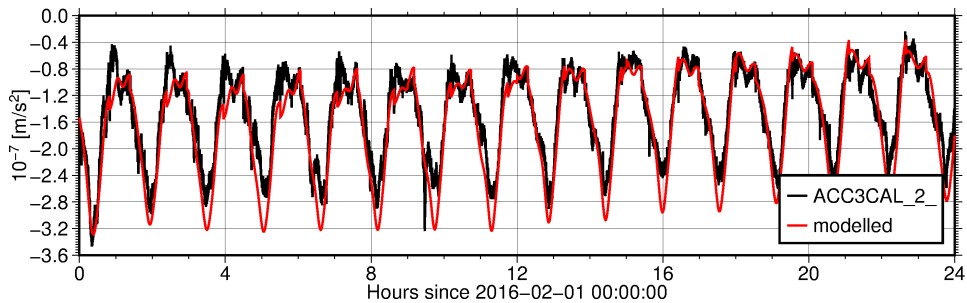

**Figure 4.** Along-track acceleration of Swarm C. The black curve shows the ACC3CAL_2_ product from Siemes et al. (2016), while the red curve shows our modelled non-gravitational accelerations (without applying any bias or scale factors).

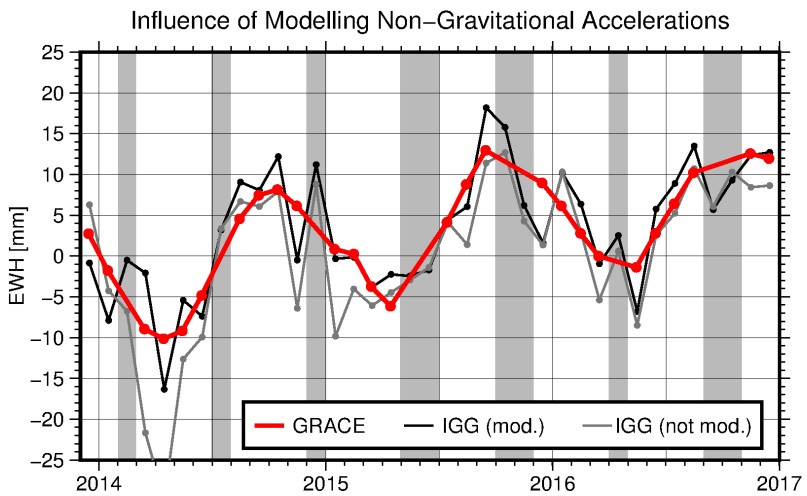

**Figure 5.** Effect of modelling of non-gravitational forces on ocean mass computation. IGG(mod.) is the monthly solution described in Table 3. The only difference in IGG(not mod.) is that non-gravitational accelerations were not modelled, but a constant value per arc was still co-estimated.

## 4.2 Effect of modelling of non-gravitational forces

Figure 4 compares modelled non-gravitational accelerations (see Sect. 3.1) to the ACC3CAL_2_ product from Siemes et al. (2016) who removed sudden bias changes from the accelerometer measurements and corrected the low-frequencies with POD-derived non-gravitational accelerations. Both time series are very close together, which supports our use of the modelled non-gravitational accelerations for gravity field estimation. Small systematic deviations can be compensated for by co-estimating additional bias or scale parameters.

Modelling non-gravitational accelerations from the Swarm satellites within TVG recovery provides an ocean mass time series significantly closer to the one from GRACE (see Fig. 5) and it also improves the trend estimate as can be seen in Table 8. This means that errors caused by neglecting non-gravitational accelerations would propagate in the spherical harmonic coefficients.

|  | GRACE | IGG | IGG (not mod.) | IGG (CTAS) |
|---|---|---|---|---|
| Trend [mm/yr] | 3.5 | 3.3 (4.3) | 4.0 (4.4) | 3.5 (3.5) |
| Amplitude (annual) [mm] | 7.9 | 6.8 (7.8) | 8.3 (9.3) | 7.4 (7.3) |
| Amplitude (semiannual) [mm] | 1.1 | 2.3 (3.2) | 2.6 (3.3) | 1.9 (1.9) |
| Phase (annual) [days] | -12.0 | -12.8 (-13.2) | -13.1 (-12.9) | -10.6 (-10.6) |
| Phase (semiannual) [days] | 6.6 | 7.8 (9.4) | 3.5 (4.9) | 8.7 (8.7) |

**Table 8.** Comparison of different IGG Swarm solutions. IGG: best monthly IGG solution. IGG (not mod.): same parameterization as IGG, but non-gravitational accelerations are not modelled. IGG (CTAS): IGG solution with an estimated constant, trend, annual and semiannual signal per spherical harmonic coefficient. The values in brackets indicate the results for the exact same months that are available for GRACE, while the values without brackets are computed from the whole Swarm time series.

## 4.3 Effect of gravity field parameterization

Figure 6 shows (1) monthly Swarm solutions compared to (2) ocean mass derived with a CTAS signal for each spherical harmonic coefficient. Obviously, the second approach fits much better to the GRACE time series, depicted in red: The RMSE decreases from 4.0 mm for (1) to only 1.7 mm for (2). Furthermore, we find a trend estimate of 3.5 mm yr$^{-1}$, which is surprisingly close to GRACE (see Table 8). In other words, directly parameterizing CTAS terms for each harmonic coefficient, instead of computing the usual monthly solutions, leads to solutions which are much closer to GRACE. The reason for this is that the estimation of CTAS terms from the whole Swarm period (Dec. 2013 to Dec. 2016) is more stable than estimating a set of spherical harmonic coefficients for each month. To our knowledge, this has not been investigated for Swarm, prior to this study.

## 4.4 Effect of different arc lengths

We investigated the effect of different arc lengths of 30 minutes, 45 minutes and 60 minutes on ocean mass estimates (see Fig. 7). The remaining parameters have been chosen according to our best results. For the CTAS approach, the solution with 30 minute arcs differs most from GRACE and the other two solutions, while 45 minute arcs provide the lowest RMSE (1.7 mm) and the best trend estimate (3.5 mm yr$^{-1}$). When considering monthly solutions, 30 minute arcs provide the best result (RMSE: 4.0 mm and trend: 3.3 mm yr$^{-1}$).

## 4.5 Effect of the parameterization of non-gravitational forces

In addition to modelling the non-gravitational forces, which are introduced in the gravity estimation process as accelerometer data, we carried out several tests, as listed in Table 4, concerning the co-estimation of "accelerometer bias and scale factors" (see Sect. 3.2). For both, Fig. 8 (a) and (b), we find that a global scaling factor per axis only has a minor influence.

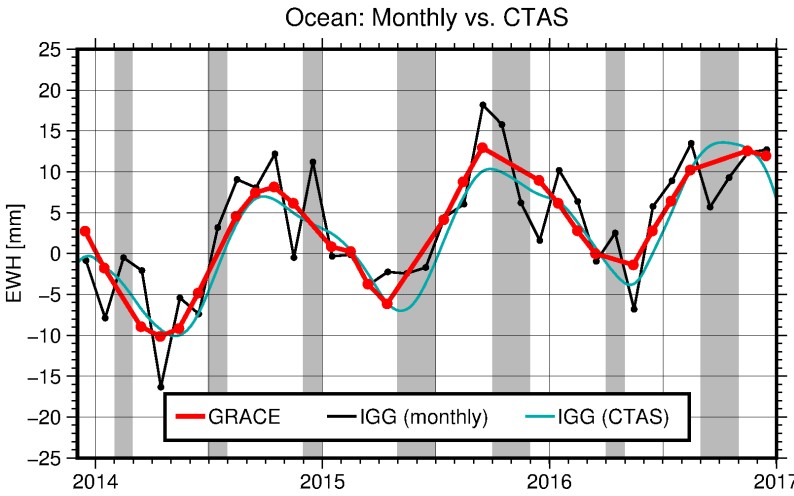

**Figure 6.** Ocean mass from GRACE and Swarm. The monthly solution is shown in black while the CTAS solution is shown in blue. The parameterizations for the two solutions can be found in Table 3.

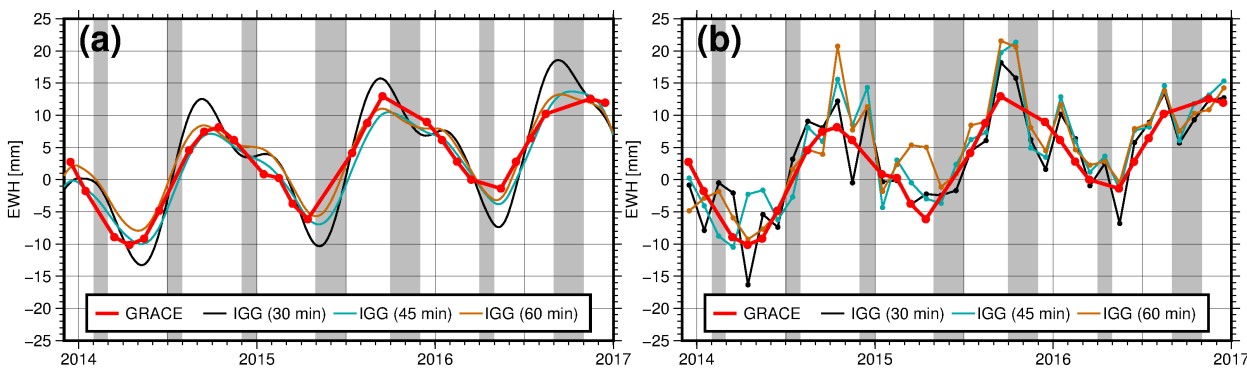

**Figure 7.** Effect of varying the arc length. (a) CTAS solution. (b) Monthly solutions.

For the CTAS solutions, parameterizing the bias as a linear function leads to a smaller RMSE with respect to the GRACE solution than a constant value per axis or not estimating it at all. The reason for this might be the large number of observations (10s sampling for 37 months) compared to the low number of parameters. The additional parameters per arc give room for improving not only the modelled non-gravitational accelerations, but also the gravity field parameters. Looking at monthly solutions, we find that a constant bias per axis has a smaller RMSE with respect to GRACE than a linear function or not estimating a bias.

For both, (a) and (b), we also introduced the bias as a constant value or a polynomial of degree 4 for the whole time span of either 37 months (a) or one month (b). The two solutions do not differ much, but they are of a minor quality compared to other solutions.

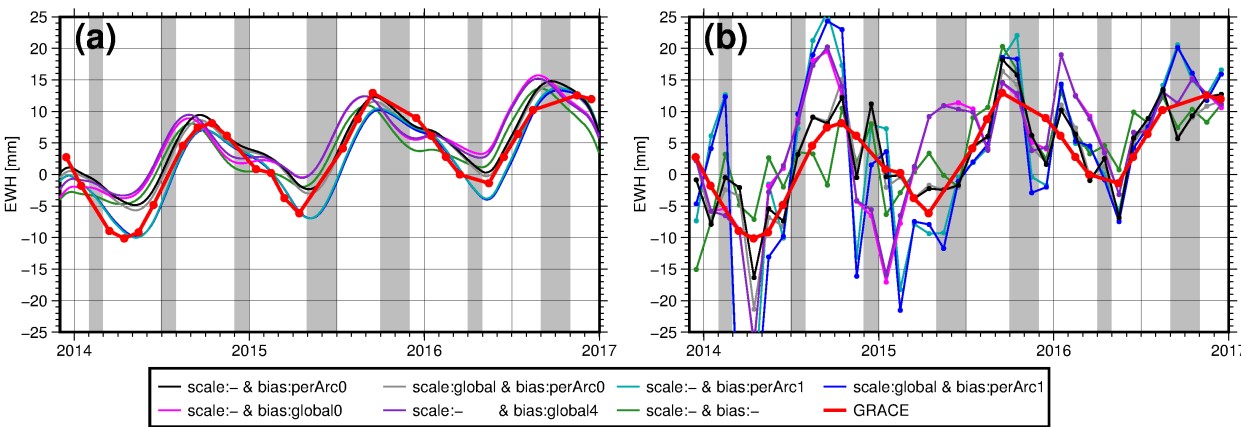

**Figure 8.** Effect of co-estimating bias and scale factors for the non-gravitational accelerations. Abbreviations: s=scale, b=bias, pA=per arc. The numbers indicate the degree of the polynomial. (a) CTAS solutions. (b) Monthly solutions.

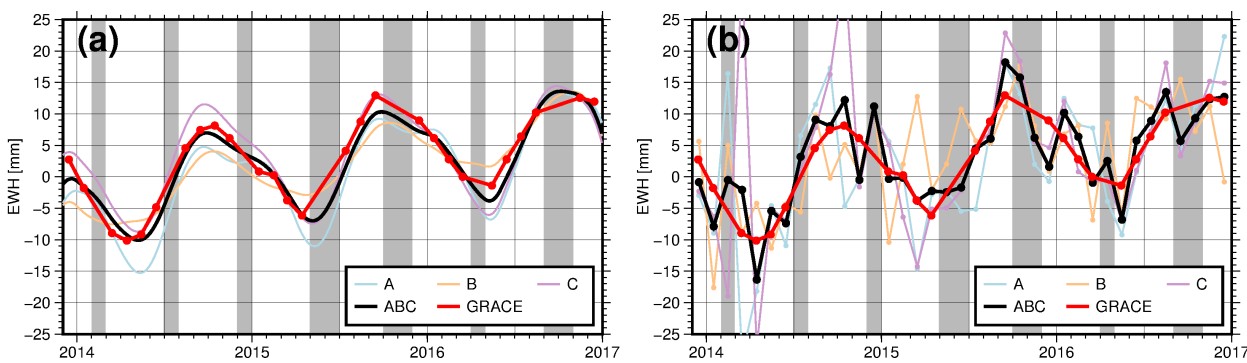

**Figure 9.** Influence of individual satellites on the combined solution. (a) CTAS solutions (b) Monthly solutions.

### 4.6 Contribution of Swarm A, B, C

In this study, we combine the information from the three spacecraft by simply accumulating the normal equations. For reasons of interpretation and validation, it makes sense to investigate also the single-satellite solutions. Figure 9 compares ocean mass change derived from the individual solutions, from the combined solution and from GRACE, for (a) the CTAS solutions and (b) the monthly solutions. It is expected that Swarm A and Swarm C provide similar solutions as they fly side by side. This is the case for the CTAS solutions, but it is not always true for the monthly solutions. One possible explanation is that the receivers have different settings, which were activated at different times (van den IJssel et al., 2016; Dahle et al., 2017).

### 4.7 River basin mass estimates

Even though we concentrated on ocean mass in this study, we also derived river basin mass estimates to validate our TVG results in land regions. We investigated the same parameterizations that we used to derive ocean mass changes (see Table 4).

To assess the solutions with regard to their quality, we compared our results to those derived from the GRACE mission. We decided to not only compute the RMSE, but to compute the ratio of the variance (VAR) of the GRACE time series to the RMSE. In this way, we can also compare the quality of the solutions in the different areas. The RMSE will be calculated with respect to the available GRACE data (27 out of 37 months from December 2013 to December 2016). This will give a kind of signal to noise ratio (SNR):

$$SNR = \frac{VAR\left(\overline{F}_{O_W, GRACE}(t)\right)}{RMSE\left(\overline{F}_{O_W, Swarm}(t)\right)} \tag{12}$$

Figure 10 shows the 100 best CTAS solutions (considering SNR for the ocean), while Fig. 11 shows an equal number of the best monthly solutions. To get an idea of the signals in the different basins, Fig. 12 shows the EWH derived from GRACE.

In general, the quality of time series of EWH derived from kinematic orbits of Swarm will be affected by (1) The basin size (see Fig. 1) and (2) the signal strength (see Fig. 12). As expected, the ratio of VAR/RMSE is lowest for the ocean, followed by the Amazon basin, which means that these results are the most reliable. The reason for the good performance of Swarm is the large basin size for the ocean and the large signal combined with a large area for the Amazon basin. For Greenland and Ganges mass estimates there exist some CTAS solutions with VAR/RMSE $> 1$ and in general, the time series of these two basins have a higher VAR/RMSE than those for the Mississippi and Yangtze basins, both for the CTAS and the monthly solutions. Considering monthly solutions, modelling non-gravitational accelerations provides better results than not modelling them. This can be seen in Fig. 11, where only very few solutions with no modelled non-gravitational accelerations are present. The best CTAS solutions for the ocean also have modelled non-gravitational accelerations, whereas for solutions 13 to 15 only empirical accelerations were co-estimated. These have been obtained with a higher VAR/RMSE for the Amazon, Mississippi, Greenland and Ganges basins. The estimation of a bias is mandatory, as both, the best CTAS and monthly solutions, always have a bias co-estimated. The best monthly solution was computed until d/o 40 and both, GRACE and Swarm were evaluated until d/o 12. This is followed by solutions that were evaluated until d/o 10. The time-variable part of the best CTAS solution is even estimated and evaluated until d/o 14. In general, the results confirm what has been evaluated in Sect. 4.2 to 4.5.

### 4.8 Bridging a possible gap with Swarm

As GRACE has met the end of its lifetime, we make efforts here to close the gap until GRACE-FO provides data. We study as well the possibility to fill monthly gaps, which are usually bridged by interpolating the previous and subsequent monthly solutions. To find out, whether Swarm TVG should be preferred to interpolating GRACE data, we assume that existing monthly solutions are missing, such that we are still able to compare to the actual solutions. In Fig. 13 (a) we assumed each individual monthly GRACE solution to be missing at one time. We then estimated a harmonic time series consisting of CTAS terms from all solutions except for the one that is considered to be missing. After having carried out the regression for each month, this leads to the blue curve. When comparing the interpolated GRACE time series to the Swarm solution, we find that they are both very close to the real GRACE solution, which offers two possibilities for bridging monthly gaps in the GRACE time series. For most months, the interpolated GRACE time series is closer to the real GRACE solution, which means that it is more reliable

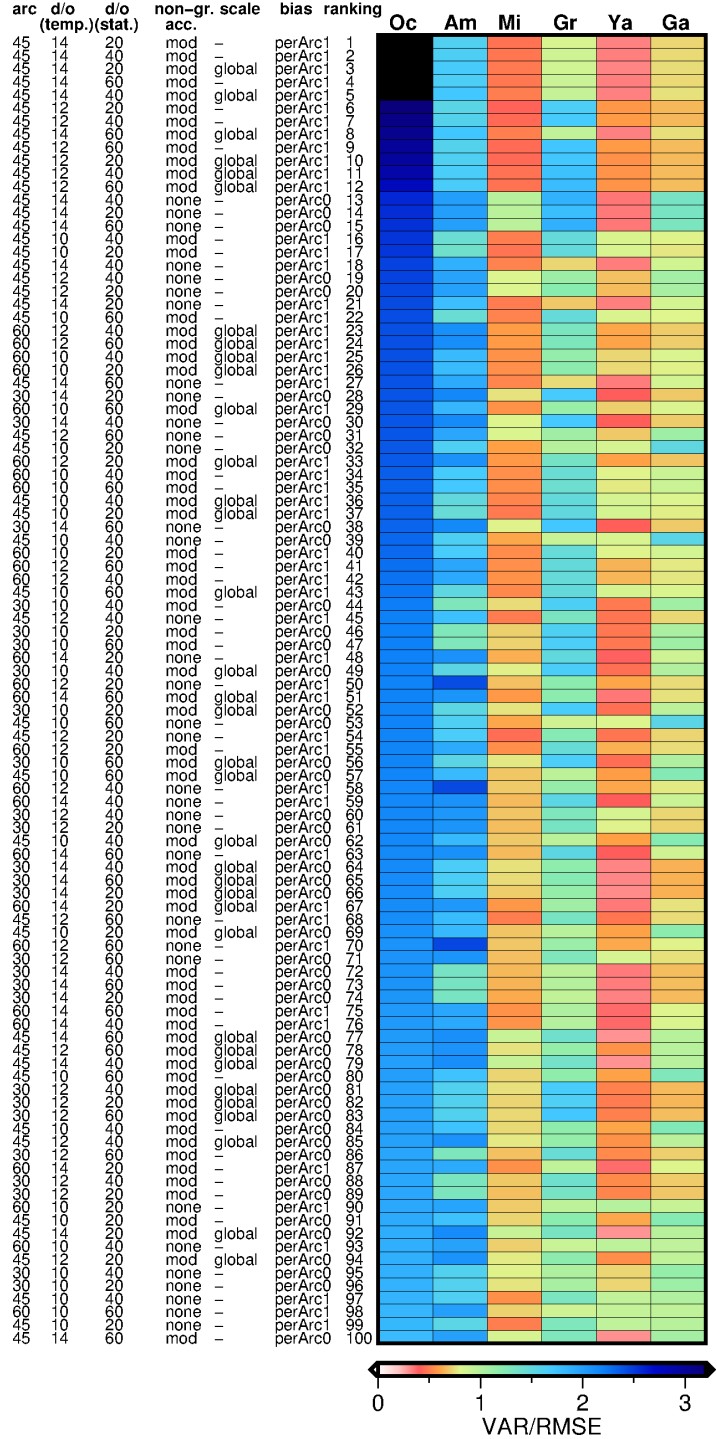

**Figure 10.** Evaluation of methods (CTAS solutions).

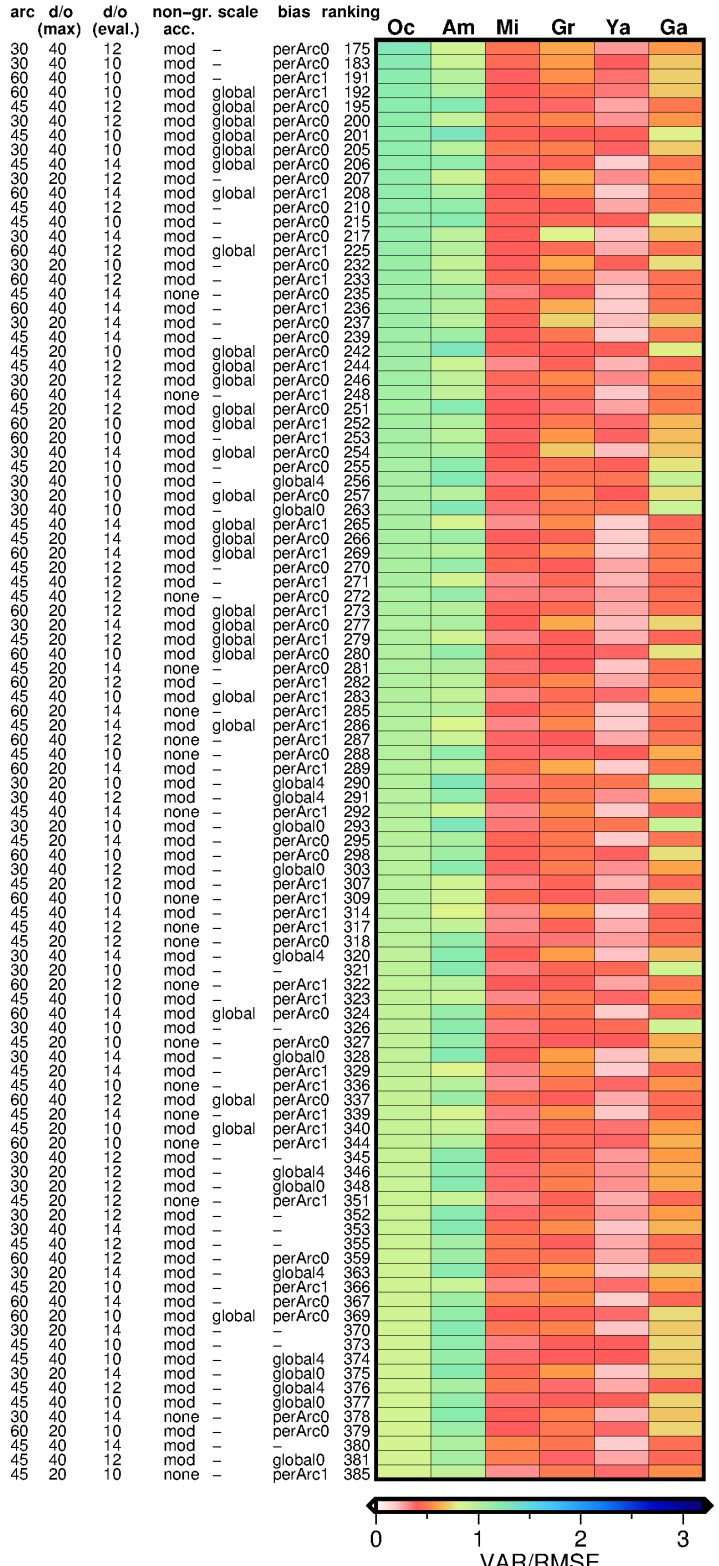

**Figure 11.** Evaluation of methods (monthly solutions).

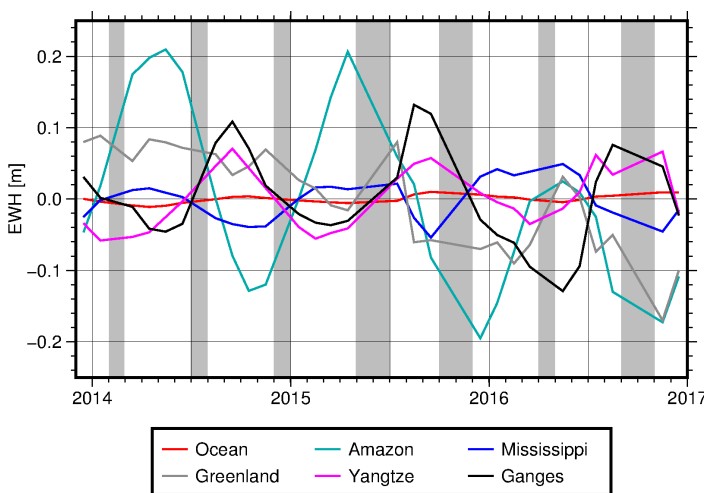

**Figure 12.** EWH derived from GRACE (d/o 12, 500 km Gaussian filter) for different regions. The time series have been reduced by their mean values for reasons of comparison.

to close monthly gaps by interpolating than by using the Swarm solutions.

In case of a longer gap between GRACE and GRACE-FO, ocean mass estimates from Swarm will become more important than considering missing monthly solutions. Figure 13 (b) shows what would happen, if the last six months of GRACE would be missing. The procedure is the same as before, i.e. we estimate a harmonic signal (CTAS) from the remaining GRACE data. This leads to the blue curve, which is further offset from GRACE than our Swarm solution. Over three years, this would also lead to a degradation of the trend estimate (GRACE and Swarm: 3.5 mm yr$^{-1}$ and interpolated GRACE: 4.0 mm yr$^{-1}$), indicating that Swarm is useful to bridge longer gaps, which will be investigated in the following.

We simulated all possible gaps with a duration between one month and 18 months in the time series from December 2013 to December 2017 and tested all gap filling methods (interpolating GRACE, using monthly Swarm solutions, using CTAS Swarm solutions). With other words: when we assumed a gap of e.g. three months, we investigated gaps from 12-2013 to 02-2014 until 10-2016 to 12-2016, which makes 35 possibilities. The mean RMSE with respect to the real GRACE data is shown in Table 9. It is obviously better to use our CTAS solution to fill gaps instead of using monthly solutions. However, for a gap of e.g. three months, we get a mean RMSE of 1.1 mm for interpolating existing GRACE solutions compared to 1.5 mm for the CTAS solution, which indicates that in most cases of a three months gap, interpolating the remaining GRACE solutions is closer to GRACE than using the Swarm solutions. For a prolonged gap of 18 months, our Swarm solution would, however, be closer to GRACE in 80 % of all cases.

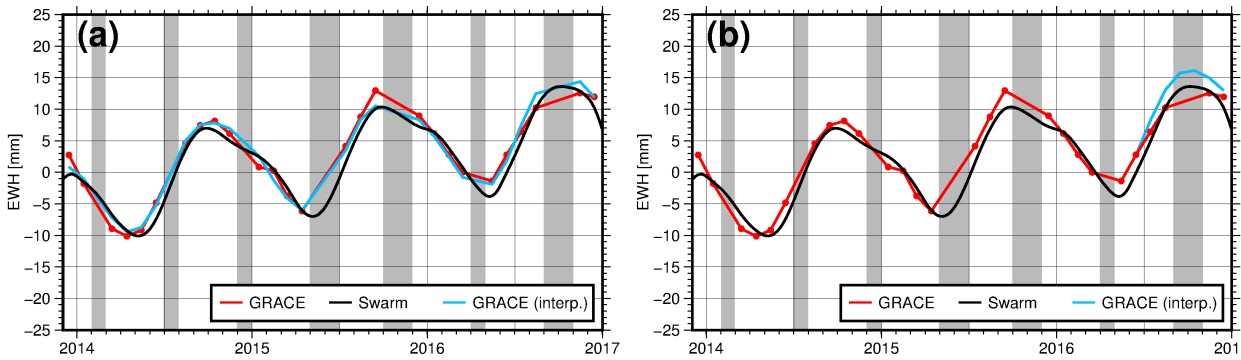

**Figure 13.** Bridging Gaps with Swarm. Our IGG Swarm solution (black) is compared to the monthly GRACE solutions (red) as well as to interpolated values when we assume a part of the GRACE time series to be missing. (a) Each month is assumed to be missing and is interpolated from all other months. (b) The last six months are assumed to be missing and are interpolated from all other months.

|  | 1 | 3 | 6 | 12 | 18 |
|---|---|---|---|---|---|
| GRACE (interpolated) | 0.9 | 1.1 | 1.1 | 1.2 | 1.8 |
| Swarm (CTAS) | 1.4 (13.5 %) | 1.5 (17.1 %) | 1.6 (6.3 %) | 1.6 (3.8 %) | 1.6 (80.0 %) |
| Swarm (monthly) | 3.3 | 3.7 | 3.8 | 3.9 | 3.8 |

**Table 9.** Mean RMSE [mm] of the gap-filler methods with respect to existing GRACE data. The columns indicate the number of missing months. The percentage of Swarm (CTAS) solutions with a lower RMSE than GRACE (interpolated) solutions is indicated in brackets. To derive the value in brackets, we counted the number of CTAS solutions with a lower RMSE than GRACE (interpolated) and computed the relation to the absolute number of CTAS solutions. The number of investigated solutions decreases from left to right, as the time span becomes longer.

### 4.9 Is it possible to detect La Niña events with Swarm?

With the Swarm accuracy as discussed in Table 6, the next logical question would be to ask what kind of sea level signal could be detected with Swarm. During the time span investigated here (Dec. 2013 to Dec. 2016), ocean mass evolves rather regularly, i.e. without apparent interannual variation. Therefore, we decided to look in the past.

5   Boening et al. (2012) and Fasullo et al. (2013) showed that the 2010/2011 La Niña event led to a 5 mm drop in Global Mean Sea Level (GMSL). This has been derived from satellite altimetry as well as from a combination of GRACE and Argo data. As most of the anomaly has been shown to be caused by mass changes, it is reasonable to ask whether we would have been able to observe the drop in ocean mass with Swarm (or to observe a similar event in the future). A simple computation tells that with an RMSE of 4.0 mm for monthly Swarm solutions, we would be able to detect a 6-months-drop of $4.0 \, \text{mm} \, / \sqrt{6} = 1.6 \, \text{mm}$.

10   As the 2010/2011 drop was both larger and lasted longer, we conclude we should have been able, and thus will probably also be able in the future, to detect La Niña events with Swarm.

We have conducted another simulation experiment with simulated ocean mass data from 1993 to 2004 taken from Wenzel

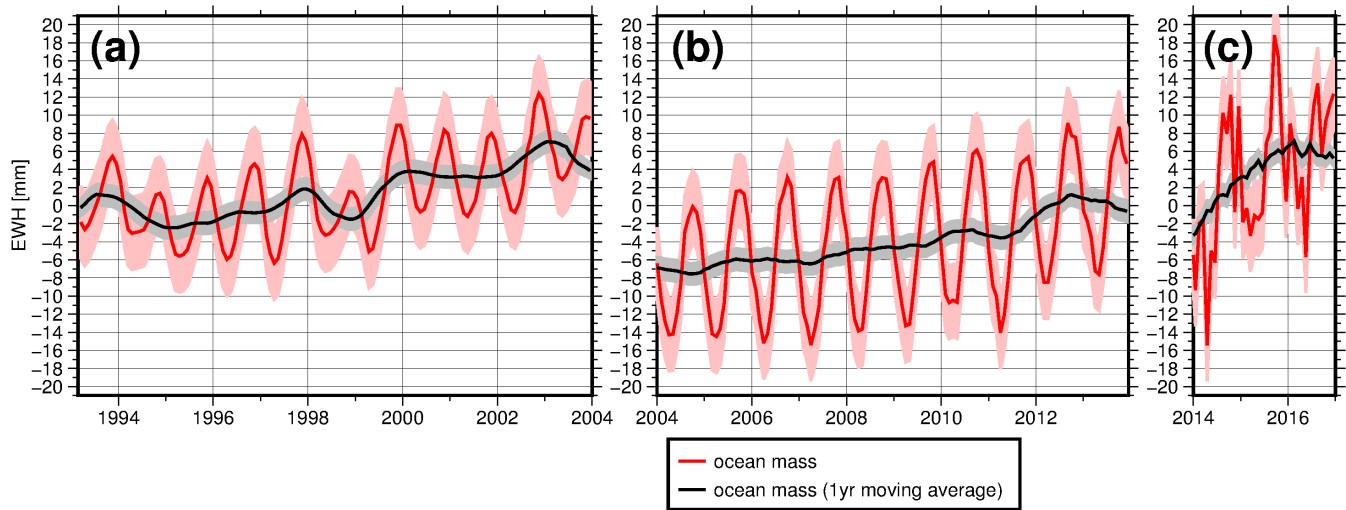

**Figure 14.** Time series of ocean mass (red). A variance of 4.0 mm is shaded in pink to simulate the uncertainty of Swarm mass estimates. A moving average filter of one year is applied (black) and the resulting standard deviation is shown in gray. (a) Simulation of ocean mass (red) from Wenzel and Schröter (2007)); 1993-2004. (b) Ocean mass from GRACE; 2004-2014. (c) Ocean mass from Swarm; 2014-2016. There is an offset between (a) and (b)/(c), because of different mean fields.

and Schröter (2007) (see Fig. 14 (a)). Assuming the Wenzel and Schröter time series as the truth here, we then generate 1000 simulated Swarm time series by adding white noise with a variance of 4.0 mm (pink area). When comparing the filtered (moving average of one year) time series shown as black line with standard deviation of 1.2 mm derived from the simulated Swarm time series (gray), we can clearly identify the drop in the years 1998 to 2000 standing out against the noise floor. Summarized, strong La Niña events such as they occurred in the past could be observed with Swarm, which will be of special importance in case of a prolonged gap between GRACE and GRACE-FO.

So far, ocean mass has been shown without adding back the GAD product from the German Research Centre for Geosciences (Flechtner et al., 2015) to our previous time series, since our focus is on comparing estimates and the GAD product has a trend of zero for the ocean basin. Here, for better interpretation, we show ocean mass from GRACE for 2004 to 2014 (Fig. 14 (b)) and ocean mass from Swarm for 2014 to 2016 (Fig. 14 (c)) with the GAD product added back.

## 5 Conclusions

Swarm-derived ocean mass estimates show the same behavior as those from GRACE, but they appear overall noisier, as expected. IGG monthly solutions have an RMSE with respect to GRACE of 4.0 mm, which is comparable or better than the solutions from other institutions that we investigated (AIUB, ASU, IfG). Over the Swarm period we find a mass trend of 3.3 mm yr$^{-1}$, which is close to that from GRACE (3.5 mm yr$^{-1}$). The spread between the different Swarm solutions is of the same order of magnitude as the RMSE of Swarm with respect to GRACE. The degree variances for monthly solutions suggest

that the TVG fields are only reliable up to about degree 10-12.

In a second approach we estimated CTAS terms for each spherical harmonic coefficient and for the whole period of time under study (Dec. 2013 to Dec. 2017). We find that this significantly improves the agreement with GRACE regarding ocean mass trend estimates; here we obtain an RMSE of 1.7 mm and the same trend as derived from GRACE. We investigated different parameterizations and found that an arc length of 30 minutes provides the best results for monthly solutions, while 45 minutes is the best option for the CTAS solutions. Furthermore, co-estimating an "accelerometer bias" proved to be important. A constant bias per arc and axis leads to the lowest RMSE with respect to GRACE for monthly solutions and an additional trend parameter is needed for the CTAS approach.

We validated TVG results by computing river basin mass estimates and comparing to GRACE. We found that the VAR/RMSE ratio, which can be considered as a signal-to-noise ratio, is highest for the ocean, followed by the Amazon basin. Some of the Greenland and Ganges solutions also show a SNR larger than one, while Swarm-derived surface mass change over the Yangtze and Mississippi is worse.

We tested three different methods for filling the gap that now will occur between GRACE and GRACE-FO, as well as for reconstructing missing single months in the GRACE time series: (1) interpolating existing monthly GRACE solutions, (2) using monthly Swarm solutions, (3) using the CTAS Swarm solution. As expected, (3) provides better results than (2) and whether (1) or (3) is better depends on the length of the gap and on the presence of episodic events and interannual variability. In the (short) Swarm period where ocean mass displayed little variability beyond the annual cycle, we found that for reconstructing either single months or three-month periods (1) may work slightly better than (3), whereas in case of a long 18 month gap, (3) should be preferred.

We showed that La Niña events like those from 2010-2011 and 1998-2000 could have been identified with Swarm, which is of special importance for the future after the termination of the GRACE mission.

In future work, we will concentrate on improving our ocean mass estimates from Swarm by allowing the trend to change over time as shown e.g. in Didova et al. (2016). Furthermore, we work towards ingesting our Swarm solutions at the normal equation level into the fingerprint inversion of Rietbroek et al. (2016), to improve existing sea level budget results and to partition altimetric sea level changes into its different components, even for those periods where we do not have GRACE data.

*Data availability.*

- The GRACE spherical harmonic coefficients that were used for comparison can be found on ftp://ftp.tugraz.at/outgoing/ ITSG/GRACE/ITSG-Grace2016/monthly/.

- The Swarm spherical harmonic coefficients from IfG Graz can be found on http://ftp.tugraz.at/outgoing/ITSG/tvgogo/ gravityFieldModels/Swarm/.

- The Swarm spherical harmonic coefficients from ASU Prague can be found on http://www.asu.cas.cz/~bezdek/vyzkum/ geopotencial/index.php.

- The employed CERES data can be found on http://ceres.larc.nasa.gov/order_data.php.

**Appendix A**

**A1**

*Competing interests.* The authors declare that they have no conflict of interest.

5 *Acknowledgements.* This study is supported by the Priority Program 1788 "Dynamic Earth" of the German Research Foundation (DFG) - FKZ: KU 1207/21-1. The authors are grateful for the Swarm macro model as well as the calibrated accelerometer data from Christian Siemes (ESA). We furthermore want to thank Christoph Dahle for sending us the Swarm gravity fields from AIUB (Bern).
We appreciate the work of Jose van den IJssel, whose kinematic orbits are available on the ESA FTP server. Thank you to Torsten Mayer-Gürr and his colleagues (IfG Graz) and Aleš Bezděk (ASU Prague) for providing their gravity solutions online.

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
