# Peer review of "Time-variable gravity fields and ocean mass change from 37 months of kinematic Swarm orbits"

_Solid Earth, 2017_

## Referee Comment (RC1) · Anonymous Referee #1 · 13 Dec 2017

GENERAL COMMENTS

The paper deals with the time-variable gravity field signal derived from GPS data of the three Swarm satellites. The main use of the obtained signal is the study of ocean mass changes, to validate the results also several other test areas are analysed (basins of large rivers). The motivation of the paper is that recently the GRACE mission has finished its operation and there are gaps in the time series of its standard monthly gravity fields, and there will be also a longer gap before GRACE-Follow-On is launched (early in 2018). The authors show that although the Swarm GPS-based gravity fields are noisier than those from GRACE, they can be successfully used to substitute the missing GRACE monthlies, especially for ocean mass. I think this is a useful paper coming at the right time.

[Figure]

The manuscript is written in a clear way and contains new results. I recommend the publication after considering my minor comments below.

SPECIFIC COMMENTS

Page-line:

2-4: Add a reference for the Swarm mission.

2-15: "orbit is at 515 km" please add an epoch, due to drag the altitude diminishes, so add something like "orbit is at 515 km at present"

2-16 of Swarm C => of Swarm B

2-32 integral equation approach/short arc: please consider adding the information of your specific inversion method to the abstract, it might useful for readers

5-1: delete "acceleration spikes due to thruster activations, and failures of automatically detecting and correcting errors". I think both these items are erroneously here. One of the reasons to include accelerometers aboard the gravity field dedicated spacecraft is to measure and record the action of the thrusters, so that it could be eliminated from further processing. I believe the CHAMP/GRACE/Swarm accelerometers should measure without any need of automatic error corrections.

5-15: The drag coefficient Cd depends on density => The drag coefficient Cd depends on composition

6-2: please add a reference or a website link for CERES

7-7, 7-8: from biases . . . from: please re-read these two lines and modify the usage of 'from'

7-7: "sampling problems with the thermosphere density model" What do you mean? One can get whatever sampling needed from the density models.

7-9: "accelerometer bias", 7-12 "accelerometer parameterization": Please add a short

explanation to readers, who are not familiar with accelerometer data processing, to describe why you like using such a terminology (even if you do not use the actual accelerometer data). Please check this whole paragraph.

9-3: IGG comes here as an abbreviation or acronym without any explanation

12-3: improved => combined

14-8, 14-9: The same comment applies here as up to 7-9, please explain the reader (or refer him up to your previous explanation), what do you mean by treatment of your non-gravitational models in a way of "accelerometer bias and scale factors".

15-2: "One possible explanation might be the different receiver settings". And what about simply a different noise realizations in individual months?

15-7: "the root mean square (RMS) of the GRACE time series" In statistics, the RMS value is equal to the mean value squared plus variance. I guess that what you mean here is only the variance. See, e.g. https://en.wikipedia.org/wiki/Root_mean_square#Relationship_to_other_statistics

19-10: "we would have been able, and thus will also be able in the future, to detect La Niña" Consider a slightly less strong statements, even if you demonstrated an example in Fig. 13, still this is only a guess. I mean, one may say "we should have been able ...".

21-6: "could have been easily identified with Swarm" the same comment as the previous one, in my opinion especially the word 'easily' is a bit exaggerated

TECHNICAL CORRECTIONS

Page-line:

1-5: gaps during => gaps occurred during

1-16: including non-dedicated satellites => including satellites non-dedicated to gravity

field study

3-table 1: van Den IJssel => van den IJssel (twice in the table)

8-6: to also => also to

21-1: by again comparing to => by comparing them to
* * *

---

## Referee Comment (RC2) · Anonymous Referee #2 · 31 Dec 2017

The article is an good study of the capabilities of the Swarm data to determine the mass change in the ocean and in a small selection of large hydrological basins, following the short-arc approach and using the kinematic orbits provided by ESA.

I find the study and its results extremely interesting, important to the geophysical community and timely given the on-going gap between the GRACE and GRACE-FO missions. For these reasons, I support its publication.

In spite of this, I suggest that a few minor issues are addressed:

- often the results are described without any physical interpretation. This has the unfortunate consequence of making the article somewhat weaker than its potential, and leaving most of the important interpretations to the reader;

[Figure]

- the article reports a large amount of results, which is very welcome and clearly shows the exhaustive nature of the study. Unfortunately, this also means the discussion needs to be proportionally detailed. I have noted a few aspects of the study that have been left out in the discussion in the annotated PDF document;

- section 4.9 is disconnected form the remaining article, it is unnecessary for the interpretation of the main results and feels as if it was a last-minute addition (with a motivation that is unclear to me); i suggest the authors consider removing it.

Additional stylistic remarks, comments, questions and suggestions are provided in the annotated PDF, attached to this review.

Please also note the supplement to this comment:
https://www.solid-earth-discuss.net/se-2017-127/se-2017-127-RC2-supplement.pdf

[Figure]

**Supplement:**

[revised manuscript text omitted]

---

## Referee Comment (RC3) · Anonymous Referee #3 · 8 Jan 2018

The article provides an assessment of Swarm gravity field determination for the determination of ocean mass changes using the short-arc approach. With the focus on ocean mass the article is complementary to previous studies on Swarm gravity field determination. Besides the usual monthly gravity field solutions also additional solutions, parametrized by trends, annual and semi-annual periodic gravity signals, are presented. The latter solutions seem to be of particular interest for Swarm due to the limited sensitivity of the GPS high-low satellite-to-satellite tracking. Moreover the article also presents detailed investigations on the optimal parametrization in the gravity recovery process and the added value of non-gravitational force models. The manuscript is clearly written and the material well presented. Only at some points a more detailed discussion would be required. I therefore recommend a minor revision based on my comments and questions that the authors may found in the annotated pdf-file.

[revised manuscript text omitted]

---

## Author Comment (AC1) · 5 Feb 2018

**Review Answers**

Christina Lück[1], Jürgen Kusche[1], Roelof Rietbroek[1], and Anno Löcher[1]

[1]Institute of Geodesy and Geoinformation, University of Bonn, Bonn, Germany

Dear Editors, dear Reviewer,

we are very grateful for your detailed corrections and annotations. We are confident that your suggestions helped to considerably improve our manuscript. We tried to respond to all comments in the best possible way. The comments are sorted by page-line, which refer to the original, uncorrected document.

5

**Reviewer 1**

**Specific Comments:**

10

**R1C1:** 2-4: Add a reference for the Swarm mission.
**Answer:** Done. Added "(Friis-Christensen et al., 2008)"

**R1C2:** 2-15: "orbit is at 515 km" please add an epoch, due to drag the altitude diminishes, so add something like "orbit is at 515 km at present"
**Answer:** Done.

**R1C3:** 2-16 of Swarm C => of Swarm B
**Answer:** Done.

20

**R1C4:** 2-32 integral equation approach/short arc: please consider adding the information of your specific inversion method to the abstract, it might useful for readers
**Answer:** Done. We added the sentence "For this aim, we use the integral equation approach with short arcs (Mayer-Gürr, 2006) to compute more than 500 time-variable gravity fields with different parameterizations from kinematic orbits." in 1-9.

25

**R1C5:** 5-1: delete "acceleration spikes due to thruster activations, and failures of automatically detecting and correcting errors". I think both these items are erroneously here. One of the reasons to include accelerometers aboard the gravity field

dedicated spacecraft is to measure and record the action of the thrusters, so that it could be eliminated from further processing. I believe the CHAMP/GRACE/Swarm accelerometers should measure without any need of automatic error corrections.

**Answer:** The reviewer is right. We deleted this part.

5 **R1C6:** 5-15: The drag coefficient Cd depends on density => The drag coefficient Cd depends on composition

**Answer:** As suggested by Reviewer 3, we included the equation for computing the drag coefficient and thus reformulated this paragraph.

**R1C7:** 6-2: please add a reference or a website link for CERES

10 **Answer:** Done. Added "(Loeb et al., 2009)" in 6-2 and the website link in the section "Data availability".

**R1C8:** 7-7, 7-8: from biases ... from: please re-read these two lines and modify the usage of "from"

**Answer:** This paragraph has been modified (see R1C10).

15 **R1C9:** 7-7: "sampling problems with the thermosphere density model" What do you mean? One can get whatever sampling needed from the density models.

**Answer:** The reviewer is right. This paragraph has been modified (see R1C10).

**R1C10:** 7-9: "accelerometer bias", 7-12 "accelerometer parameterization": Please add a short explanation to readers, who are
20 not familiar with accelerometer data processing, to describe why you like using such a terminology (even if you do not use the actual accelerometer data). Please check this whole paragraph.

**Answer:** We modified this paragraph and wrote: "As described in Sect. 3.1 we derive non-gravitational accelerations from models, which we then use in the gravity field estimation as a proxy for accelerometer measurements. Due to the presence of errors, e.g. caused by uncertainties in the density model or errors in the macro model, the resulting non-gravitational accelera-
25 tions might not always reflect the truth. To prevent these errors from propagating into the gravity field estimates, it is common to introduce additional parameters. Here we co-estimate an "accelerometer bias" per arc and per axis, ..."

**R1C11:** 9-3: IGG comes here as an abbreviation or acronym without any explanation

**Answer:** IGG is now explained as "Institute of Geodesy and Geoinformation".

30

**R1C12:** 12-3: improved => combined

**Answer:** This sentence is now corrected as suggested in R2C52.

**R1C13:** 14-8, 14-9: The same comment applies here as up to 7-9, please explain the reader (or refer him up to your previous
35 explanation), what do you mean by treatment of your non-gravitational models in a way of "accelerometer bias and scale factors".

**Answer:** Thank you. We referred the reader to our new explanation in Sect. 3.2.

**R1C14:** 15-2: "One possible explanation might be the different receiver settings". And what about simply a different noise realizations in individual months?

**Answer:** We reformulated this sentence: "One possible explanation is that the receivers have different settings, which were activated at different times (van den IJssel et al., 2016; Dahle et al., 2017)". We found that the differences in the monthly solutions appear bigger than the noise level would suggest.

**R1C15:** 15-7: "the root mean square (RMS) of the GRACE time series" In statistics, the RMS value is equal to the mean value squared plus variance. I guess that what you mean here is only the variance. See, e.g. https://en.wikipedia.org/wiki/Root_mean_square#Relationship_to_other_statistics

**Answer:** The reviewer is right. We thank him/her for this correction. The computations were conducted with the variance. Table 6 on page 10 was however the RMS. We decided to compute the variance instead, as this is what we actually wanted.

**R1C16:** 19-10: "we would have been able, and thus will also be able in the future, to detect La Niña" Consider a slightly less strong statements, even if you demonstrated an example in Fig. 13, still this is only a guess. I mean, one may say "we should have been able ...".

**Answer:** We changed the sentence to "we should have been able, and thus will probably also be able in the future, to detect La Niña"

**R1C17:** 21-6: "could have been easily identified with Swarm" the same comment as the previous one, in my opinion especially the word "easily" is a bit exaggerated

**Answer:** We removed the word "easily".

**Technical Corrections:**

**R1C18:** 1-5: gaps during => gaps occurred during
**Answer:** Done.

**R1C19:** 1-16: including non-dedicated satellites => including satellites non-dedicated to gravity field study
**Answer:** Changed to: "including satellites not dedicated to gravity field studies"

**R1C20:** 3-table 1: van Den IJssel => van den IJssel (twice in the table)

**Answer:** Thank you. We corrected this mistake.

**R1C21:** 8-6: to also => also to

**Answer:** This sentence has been reformulated to "We employed an ocean mask that includes the Arctic ocean and does not have a coastal buffer zone." (see R2C43).

**R1C22:** 21-1: by again comparing to => by comparing them to

**Answer:** Done (also left out "by" as suggested in R2C100).

---

## Author Comment (AC2) · 5 Feb 2018

**Review Answers**

Christina Lück[1], Jürgen Kusche[1], Roelof Rietbroek[1], and Anno Löcher[1]

[1]Institute of Geodesy and Geoinformation, University of Bonn, Bonn, Germany

Dear Editors, dear Reviewers,

we are very grateful for your detailed corrections and annotations. We are confident that your suggestions helped to considerably improve our manuscript. We tried to respond to all comments in the best possible way. The comments are sorted by page-line, which refer to the original, uncorrected document.

**Reviewer 2**

**General Comments:**

**R2C1:** often the results are described without any physical interpretation. This has the unfortunate consequence of making the article somewhat weaker than its potential, and leaving most of the important interpretations to the reader.

**Answer:** Indeed, the focus of this paper is on the various analysis options for Swarm data and on the variation of Swarm ocean mass with respect to GRACE. It is a technical paper, therefore we will write a paper on Swarm ocean mass and its physical

15 interpretation once we have analyzed the current period of data where no GRACE truth is present.

**R2C2:** the article reports a large amount of results, which is very welcome and clearly shows the exhaustive nature of the study. Unfortunately, this also means the discussion needs to be proportionally detailed. I have noted a few aspects of the study that have been left out in the discussion in the annotated PDF document

20 **Answer:** Thank you. We will try to accommodate missing details in our revision.

**R2C3:** section 4.9 has been disconnected form the remaining article, it is unnecessary for the interpretation of the main results and feels as if it was a last-minute addition (with a motivation that is unclear to me); i suggest the authors consider removing it.

25 **Answer:** The reviewer is right that Sect. 4.9 is disconnected from the remaining article. However, we feel that these are important results and we now tried to motivate the section in the introduction as well as in the beginning of Sect. 4.9. See 2-2: "As we will see in the course of this paper, the European Space Agency (ESA) Swarm Earth Explorer mission (Friis-Christensen et al.,

2008) is able to detect regular as well as non-regular ocean mass changes such as La Niña events." and 19-5: "With the Swarm accuracy as discussed in Table 6, the next logical question would be to ask what kind of sea level signal could be detected with Swarm. During the timespan investigated here (Dec. 2013 to Dec. 2016), ocean mass evolves rather regularly, i.e. without apparent interannual variation. Therefore, we decided to look in the past".

**R2C4:** Additional stylistic remarks, comments, questions and suggestions are provided in the annotated PDF, attached to this review.

**Answer:** Thank you very much for your detailed annotations. The answers to your comments can be found below. We hope the flow of the story is better now.

**Specific Comments:**

**R2C5:** 1-1: replace "one to partition" with "for partitioning"
**Answer:** Done.

**R2C6:** 1-2, 1-3: replace "..., into a mass-driven and a steric part, the latter being related to ocean heat change and the current Earth's energy imbalance" with "..., into mass-driven and steric parts. The latter is related to ocean heat change and the Earth's current energy imbalance"
**Answer:** We included the changes, except that we kept "Earth's energy imbalance", since this is a fixed term (EEI).

**R2C7:** 1-5: "there will be a prolonged gap": "appreciable" The first models from GRACE-FO are expected to be produced around July 2018, so it is arguable if 6 months is a "prolonged" gap. It is certainly something that cannot be ignored.
**Answer:** The reviewer is absolutely right. The gap will be six months or longer. For clarity we stick to "prolonged gap". But maybe we misunderstood the suggestion?

**R2C8:** 1-12: replace "our Swarm solution" with "our trend, annual and semiannual Swarm solution"
**Answer:** Done. We decided to write "our CTAS Swarm solution" to be consistent with the remaining article. CTAS is explained as "constant, trend, annual and semiannual" at its first occurence.

**R2C9:** 1-13: add ", when artificially removing one solution." behind "appears better than interpolating existing GRACE data in 13.5 % of all cases"
**Answer:** Done.

**R2C10:** 1-13, 1-14: replace "..., for 80.0 % of all investigated cases of an 18-months-gap, Swarm ocean mass was found..." with "In case of an 18-months artificial gap, 80.0 % of all trend, annual and semiannual Swarm solutions were found..."

**Answer:** Done.

**R2C11:** 2-5 to 2-7: "For this aim, the satellites are equipped with absolute scalar and vector field magnetometers. Furthermore, a suprathermal ion imager and a Langmuir Probe provide information about the Earth's electric field." Although these state-
5  ments are correct, they are not relevant to this article. Consider removing them.

**Answer:** Done.

**R2C12:** 2-12, 2-13: The authors seem to argue that the calibrated and corrected accelerometer measurements in the along-track direction derived by Siemes et al. 2016 are not useful to gravity field determination. Please add a few statements supporting
10  this argument.

**Answer:** We do not argue that the calibrated and corrected accelerometer measurements in the along-track direction derived by Siemes et al. 2016 are not useful to gravity field determination. In fact, it is very useful to have knowledge about the non-gravitational accelerations. We just stated that we do not have calibrated and corrected accelerometer measurements for Swarm A and B as well as for across-track and radial directions (and, as far as we know, we will probably not get them in the future).

**R2C13:** 2-15, 2-16: replace "Swarm C" with "Swarm B"

**Answer:** Done.

**R2C14:** 2-16: replace "with respect to the other two planes" with "with respect to the orbital planes of the other two satellites"
20  **Answer:** Done.

**R2C15:** 2-17: replace "basis" with "opportunity"

**Answer:** Done.

25  **R2C16:** 2-20, 2-21: "At the time of writing, kinematic LEO orbits are considered as a promising option for deriving global gravity fields during a GRACE mission gap (Rietbroek et al., 2014)": Many authors have argued about this possibility before Rietbrok et al. 2014, e.g.: Gunter, B. C., Encarnação, J., Ditmar, P., & Klees, R. (2009). The use of satellite constellations and formations for future gravity field missions. In Advances in the Astronautical Sciences (pp. 1357-1368). Savannah. Retrieved from http://www.univelt.com/book=1451 Weigelt, M., Van Dam, T., Jäggi, A., Prange, L., Tourian, M. J., Keller, W., Sneeuw,
30  N. (2013). Time-variable gravity signal in Greenland revealed by high-low satellite-to-satellite tracking. Journal of Geophysical Research: Solid Earth, 118(7), 3848-3859. http://doi.org/10.1002/jgrb.50283

**Answer:** Thank you for these suggestions, we included the two references.

**R2C17:** 2-28: The correct reference to these solutions is (so far): Zehentner, N. (2016). Kinematic orbit positioning applying
35  the raw observation approach to observe time variable gravity. Graz University of Technology. Retrieved from https://www.

researchgate.net/publication/316668578_Kinematic_orbit_positioning_applying_the_raw_observation_approach_to_observe_time_variable_gravity

**Answer:** Thank you, we added the reference.

5   **R2C18:** 2-28: insert "monthly" in "...suggesting that a meaningful *monthly* time-varying gravity signal..."
**Answer:** Done.

**R2C19:** 2-29: add "..., considering the average of the three models."
**Answer:** Done.

**R2C20:** 3-15: Please discriminate between earth albedo radiation (the solar radiation reflected by the earth's surface) and the thermal radiation energy (the component resulting from Earth's internal heat). Does this model distinguish land and ocean surface?

**Answer:** We added: "and Earth radiation pressure consisting of measured albedo and emission". CERES data are available as
15   gridded data (over land and ocean).

**R2C21:** 3-20: "ITSG Graz solutions": I believe the Geodesy group in Graz changed its name to IfG (Institut für Geodäsie), please confirm.

**Answer:** The reviewer is right. The GRACE solutions that we use to compare are however still called "Graz ITSG-Grace2016",
20   so we reformulated the sentence to "...were chosen such as to be aligned with the Graz ITSG-Grace2016 solutions". When we talk about the IfG Swarm solutions, we changed from "ITSG" to "IfG".

**R2C22:** 4-Table 2: Does this mean that no relativistic effects have been considered?

**Answer:** The reviewer is right. Relativistic effects have not been considered here. Their impact on the gravity fields would be
25   negligible. However, we still plan to implement this in our further studies.

Eshagh and Najafi (2007): Perturbations in orbital elements of a low earth orbiting satellite. Journal of the Earth & Space Physic s. Vol. 33, No. 1

give an overview on how orbits of LEO satellites are affected by tidal forces, non-gravitational forces and relativistic effects. The relativistic part is always considerably lower than the others.

30

**R2C23:** 4-6, 4-7: "In Vielberg et al. (subm.) we compare NRLMSISE-00 to GRACE-derived thermospheric density and derive an empirical correction for this model; this has not yet been applied here.": I fail to see the motivation to make this statement, please clarify.

**Answer:** Since Vielberg et al. is submitted to the same special issue and some computations are similar to the ones in this paper

(concerning non-gravitational accelerations) we thought it would be good to mention in order to avoid misunderstandings.

**R2C24:** 5-1: "and failures of automatically detecting and correcting errors": It is not clear to me what the authors mean with this statement. Possibly it is connected with the so-called Error Detection And Correction (EDAC) failure events which is an exceptional but expected behavior of the on-board computer, when internal operational checks determine that a memory failure can only be recovered after a reboot.

**Answer:** The reviewer is right. This part was confusing and we deleted it.

**R2C25:** 5-2 to 5-4: "However, Swarm A and B, as well as the other C directions have stronger problems and it is not clear whether these data can be used in the future.": (1) add "(the former to a less extent than the latter)" after "However, Swarm A and B", (2) I would argue that a better reason not to use Swarm-C accelerometer data for this type of gravity field studies is the fact that the data is has large gaps, of varying quality, and published with a long and unspecified latency. The authors are evidently free to decide not to tackle the issue of determining the usefulness of these data to gravity field studies; but is the author's choice, not the result of the how clear the issue is. (3) replace "in the future" with "gravity field applications" (the Swarm-C data has been used to study the neutral atmospheric response to solar storms, so, formally, the current statement is not correct)

**Answer:** (1) Done. (2) We think this is a misunderstanding. We do not talk about the Swarm C along-track data, but about all other accelerometer data that will probably not be used in the future. (3) Done.

**R2C26:** 5-14: "$A_{ref}$ is the surface area of the spacecraft": Does this mean that the 15 macro model mentioned in p. 3 l. 14 is not used for atmospheric drag? If so, please explicitly state so and motivate this inconsistency.

**Answer:** We use the macro model with its 15 panels for the computation of the acceleration due to drag. The drag coefficient Cd needs to be computed for all 15 panels. We explained this in more detail and refer to Doornbos 2011, as this is a rather long computation.

**R2C27:** 5-16: add a comma: "The drag coefficient Cd depends on density, temperature and the macro model properties"," and we follow Doornbos (2011) in its computation."

**Answer:** We reformulated this paragraph, so that this correction was not neccessary anymore.

**R2C28:** 5-22: replace "accumulates SRP for" with "accounts for the SRP over"

**Answer:** Done.

**R2C29:** 6-20: "varying low degrees": Does this mean the maximum degree changes from solution to solution? Please clarify.

**Answer:** We tested different maximum d/o, but it does not change from month to month. This should hopefully be clear from Table 4. This is now also clarified later in the text: "As Fig. 3 only shows the degree variances for one particular month, we

investigate different maximum degrees in the following (see Table 4). We evaluate our monthly fields until d/o 10, 12 or 14. Even though higher degrees do not contribute a reasonable time-variable signal, we estimate the monthly fields until d/o 20 or 40, because high degrees can absorb errors that would otherwise propagate in the lower degrees. For our CTAS solution, we estimate a static part ($\overline{c}_{nm}$ and $\overline{s}_{nm}$ in Eq. 9) until d/o 20, 40 or 60, while the time-variable part is estimated until d/o 10, 12 or 14."

**R2C30:** 6-21: "In a single adjustment, a set of trends and (semi)annual harmonic amplitudes are additionally estimated for each Stokes coefficient, which is likely more reasonable when aiming at a long and stable time series": This word is confusing. Do the authors suggest that monthly gravity fields are not reasonable for Swarm?

**Answer:** We do not suggest that monthly gravity fields are not reasonable for Swarm. We just suggest that fitting a harmonic function to each spherical harmonic coefficient might be more reasonable for our purpose of creating a long and stable time series. We reformulated the sentence: "As we aim at a long and stable time series, we additionally parameterize a set of trends and (semi)annual harmonic amplitudes to the constant part for each Stokes coefficient in a single adjustment".

**R2C31:** 7-9: "accelerometer bias": Is there a reason for the quotation marks?

**Answer:** We put the term in quotation marks, because we do not use accelerometer data, but our modelled non-gravitational accelerations, which are treated as accelerometer measurements. We still wanted to use the fixed term, so we put it in quotation marks. As this might have been confusing, we rewrote the paragraph:

"As described in Sect. 3.1 we derive non-gravitational accelerations from models, which we then use in the gravity field estimation as a proxy for accelerometer measurements. Due to the presence of errors, e.g. caused by uncertainties in the density model or errors in the macro model, the resulting non- gravitational accelerations might not always reflect the truth. To prevent these errors from propagating into the gravity field estimates, it is common to introduce additional parameters. Here we co-estimate an "accelerometer bias" per arc and per axis, ..."

**R2C32:** 7-9: "low-degree polynomial": Is there a reason not to state the degree explicitly?

**Answer:** We reformulated: "... either as a constant value or with an additional trend parameter"

**R2C33:** 7-11: "Another possibility would be...": This phrase does not make it clear if this possibility is going to be evaluated in this article or not.

**Answer:** We reformulated the sentence to: "Another possibility that is also evaluated in this paper is ..."

**R2C34:** 7-11: "higher degree": Is there a reason not to state the degree explicitly?

**Answer:** This part has now been deleted, as suggested in R3C43.

**R2C35:** 7-13: "The influence of this "accelerometer parameterization" will be evaluated in the course of this paper, yet one needs to bear in mind that these parameters rather measure force model inconsistencies and should not be mixed up with instrument errors": This statement is certainly true but it seems to come out of nowhere. It is not clear to me the reason a reader would at this point make this confusion.

**Answer:** We decided to leave the sentence like this. It should be clear to the reader that we account for errors in our modelling of non-gravitational accelerations and that we do not use accelerometer data (see R2C31).

**R2C36:** 7-23: "with the smoothing Kernel W , here a 500 km Gaussian filter": I suggest placing this statement before Eq. 9 and adapting that sentence, e.g. "The smoothed region average F, considering the smoothing kernel W (here 500km Gaussian), over the region O can be expressed as"

**Answer:** Done.

**R2C37:** 7-1: replace "may or may not replace" with "test replacing"

**Answer:** Done.

**R2C38:** 8-Table 3: (1) Is there a reason for the different default arc lengths for the monthly and trend+annual+semiannual cases? (2) Is there a reason for the different default bias parametrization for the monthly and trend+annual+semiannual cases? (3) "static until 40": To be clear, is this related to the constant terms in Eq. 8? So in that equation, the subscripts nm have different maximum values for the constant terms? (4) Also, formally, the name trend+annual+semiannual should be constant+trend+annual+semiannual. Probably an acronym would be more readable, e.g. CTAS (or TAS, should the authors prefer the current term).

**Answer:** (1) and (2): We chose these parameterizations because they represent our best monthly solution (as will be seen in Fig. 11) and the best CTAS solution up to degree and order (d/o) 12 (see Fig. 10). The choice of the same degree allows a comparison of the results. (3) Yes, the explanation is correct. This is now clarified as explained in R2C29. (4) Thank you for the suggestion to use CTAS, we changed the name in the whole document.

**R2C39:** 8-1: "While replacing $c_{20}$ leads to a workflow more in line with GRACE...": Please motivate this advantage, it is not clear to me why setting some processing details to match those of GRACE is an advantage (I would say it is not, it is merely a choice that seems to produce more realistic results). Possibly the authors wish to compare their results with GRACE and this choice makes that comparison more straightforward. Still, that comparison should be motivated.

**Answer:** We made this statement because it is the usual procedure to replace the $c_{20}$ coefficient, as it is done in GRACE. We however investigate the question whether Swarm alone is able to measure mass changes. But we would like to avoid, here, a discussion on which $c_{20}$ estimate is more realistic.

**R2C40:** 8-3: "we substitute all degree 1 coefficients": I deduce from this statement that the degree 1 coefficients are co-estimated along with the remaining higher-degree coefficients. In my experience (considering a different approach), this leads to higher errors in degrees above 1, as the power contained i those degrees aliases to the degree 1 coefficients (which are not

well observed). It would be reassuring if the authors tested estimating the gravity field without co-estimating degree 1 coefficients and check if this modifications produces significantly different results (in case the authors have not yet done this).

**Answer:** The reviewer is right. Co-estimating degree 1 coefficients would probably lead to problems in degrees above 1. We did *not* estimate degree 1 coefficients, however, this was not clear from the discussion paper. To clarify, we replaced "substitute" with "add" and we added the sentence "We estimate the spherical harmonic coefficients from degree 2 onward." in 7-7.

**R2C41:** 8-3: "In a next step, we substitute all degree 1 coefficients to correct for geocenter motion (Swenson et al., 2008), which cannot be detected with the current GRACE and Swarm processing.": The authors should be aware that this statement ("current") raises some tricky questions. Inferred from it is the issue of whether \*any\* satellite gravimetric mission can measure the degree 1 coefficients, which, to my understanding, is not yet certain.

**Answer:** The reviewer is right, this is a tricky question. Recent research suggests that simultaneously analyzing LEO orbits in GNSS processing may help to improve degree 1 estimates. Since this is not the topic here, this is not further discussed.

**R2C42:** 8-5: "but as long as we apply the same correction to GRACE our results are independent of this choice": I disagree, the authors' results will be different independently of what is done in the case of GRACE. Possibly the authors mean that it would not be possible to see this additional processing step when comparing their results with GRACE (after doing the same in this case), but this is a rather trivial statement: the same would happen if the authors subtracted a random but constant quantity from the both theirs and GRACE coefficients.

**Answer:** The reviewer is right. We mean that the *comparison* will be independent of the correction for glacial isostatic adjustment. We changed the sentence to "... as long as we apply the same correction to GRACE, the comparison between Swarm and GRACE will be independent of this choice". We still do the correction because it is neccessary for the computation of ocean mass.

**R2C43:** 8-6: "coastal buffer zone": Please specify the width of this buffer zone.

**Answer:** This sentence was confusing. It was meant in a way that we chose an ocean mask excluding (i.e. without) a coastal buffer zone. We restrucered the sentence to: "We employed an ocean mask that includes the Arctic ocean and does not have a coastal buffer zone."

**R2C44:** 8-7: "Our test studies include all possible combinations of the parameterizations shown in Table 4, which leads to more than 500 configurations": This is a interesting and welcome study but it now seems somewhat poorly motivated earlier in the article. I also suggest that such large amount of work should be given a more noticeable place in the article (which may naturally arise from some words on its motivation).

**Answer:** This information is now also included in the abstract: "For this aim, we use the integral equation approach with short arcs (Mayer-Gürr, 2006) to compute more than 500 time-variable gravity fields with different parameterizations from kinematic

orbits".

**R2C45:** 9-Table 4: "maximum d/o": There is no discussion regarding this parameter in the text. If it is something the authors do not wish to discuss, I suggest they pick the best scenario and simply state that it was determined to be the best one in an "offline" analysis.

**Answer:** We now explained our choice of maximum d/o as explained in R2C29. In the discussion of Figure 10 and 11 we wrote: "The best monthly solution was computed until d/o 40 and both, GRACE and Swarm were evaluated until d/o 12. This is followed by solutions that were evaluated until d/o 10. The time-variable part of the best CTAS solution is even estimated and evaluated until d/o 14.".

**R2C46:** 9-1: "4.1 Ocean mass from GRACE and Swarm": When comparing monthly Swarm solutions with GRACE, how did the authors handled the fact that the GRACE monthly solutions are not strictly limited to calendar months? Are the Swarm solutions computed with the same days as the GRACE solutions? Are the Swarm solutions interpolated?

**Answer:** Our Swarm solutions are indeed strictly limited to calendar months. The reviewer is right, when comparing to GRACE, this is not optimal. We still chose to compare the solutions directly as is done in other studies.

**R2C47:** 9-2: "TVG": Please define the acronyms the first time they are used, irrespective of how obvious they are (possibly also for other acronyms).

**Answer:** Done.

**R2C48:** 9-9, 9-10: "and is still very close to GRACE": I suggest not using weak statements such as "very close", it does not bring any new information.

**Answer:** We removed this statement.

**R2C49:** 9-20: "The difference (dotted gray line) indicates that Swarm is only reliable for degrees up to about 12.": I disagree, the figure clearly shows that the degree variance increases (visually) from degree 10, which is indicative of noise and not signal. Recall that the results of Teixeira da Encarnação (2016) considered the average of 3 models, which may be the cause for this discrepancy.

**Answer:** The reviewer is right. The d/o also depends on the month, so we now wrote: "The difference (dotted gray line) indicates that for this particular month Swarm is only reliable for degrees up to about 10".

**R2C50:** 9-21: "The formal errors (dotted black line) appear to be too optimistic...": Unless calibrated, formal errors are always too optimistic. Possibly a better statement reflects this, e.g.: "Since the formal errors (dotted black line) are not calibrated, they are too optimistic..."

**Answer:** Done.

**R2C51:** 12-Figure 4: From the text and from the high-frequency content, "POD" is not the best legend for the black line. I suggest "Siemes et al. (2016)" or the "ACC3CAL_2_"

**Answer:** We replaced "POD" with "ACC3CAL_2_" and modified 12-2 and 12-3 to "Figure 4 compares modelled non-gravitational accelerations (see Sect. 3.1) to the ACC3CAL_2_ product from Siemes et al. (2016) who ..."

**R2C52:** 12-3: "... Siemes et al. (2016) who corrected the accelerometer measurements and improved them with POD": "Improved" might not be the best word. I suggest "corrected the low-frequencies"

**Answer:** We reformulated the sentence to "... who removed sudden bias changes from the accelerometer measurements and corrected the low-frequencies ..." since the word "corrected" would have been repeated.

**R2C53:** 12-3: add "-derived non-gravitational accelerations"

**Answer:** Done.

**R2C54:** 12-3, 12-4: replace "confirms our intention to use the" with "supports the/our use of"

**Answer:** Done.

**R2C55:** 12-7: "Modelling non-gravitational accelerations from the Swarm satellites within TVG recovery provides an ocean mass time series significantly closer to the one from GRACE (see Fig. 5) and it also improves the trend estimate" The authors provide these two observations but do not provide any interpretation from these observations. Can the authors provide an explanation for these observations?

**Answer:** The reviewer is right. We added the sentence "This means that errors caused by neglecting non-gravitational accelerations would propagate in the spherical harmonic coefficients.". It makes sense to emphasize this, as there were doubts when this was evaluated for CHAMP earlier.

**R2C56:** 13-3: "coefficient": Since it is an important parameter for the discussion in this paragraph, I suggest stating (again) the maximum degree up to which this estimation was done.

**Answer:** The parameterization for the IGG (monthly) and the IGG (CTAS) solution is shown in Table 3. We now point this out in the caption of Figure 6.

**R2C57:** 13-6: "... directly parameterizing trend, annual and semiannual terms for each harmonic coefficient, instead of computing the usual monthly solutions, leads to solutions which are much closer to GRACE...": Can the authors provide an explanation for this observation?

**Answer:** We added: "The reason for this is that the estimation of CTAS terms from the whole Swarm period (Dec. 2013 to

Dec. 2016) is more stable than estimating a set of spherical harmonic coefficients for each month."

**R2C58:** 14-5: "For the approach with trend, annual and semiannual signal terms, the solution with 30 minute arcs differs most from GRACE and the other two solutions, while 45 minute arcs provide the lowest RMSE (1.7 mm) and the best trend estimate

5   (3.5 mm yr$^{-1}$)": Can the authors provide an explanation for this observation?

**Answer:** R2C58 and R2C59: We are not sure about the reasons for the different arc lengths of the best monhtly and CTAS solutions. It could be related to the ratio of the number of observations to the number of unknown parameters. Both differ extremely for the two cases.

10  **R2C59:** 14-6: "When considering monthly solutions, 30 minute arcs provide the best result": Can the authors provide an explanation for this observation?

**Answer:** See R2C58.

**R2C60:** 14-9: add ", as listed in Table 4" to "several tests"

15  **Answer:** Done.

**R2C61:** 14-9: "accelerometer bias and scale factors": It is not clear to me the reason to put this phrase within quotation marks.

**Answer:** See R2C31.

20  **R2C62:** 14-11: "For the solutions with trend, annual and semiannual signal terms, parameterizing the bias as a linear function works better than a constant value per axis": Please try to use stronger statements. "Works better" can mean numerous things.

**Answer:** We reformulated this: "leads to a smaller RMSE with respect to the GRACE solution".

**R2C63:** 14-12: "large number of observations (10s sampling for 37 months) compared to the low number of parameters": If

25  this is the case, then one might question why stop at the linear calibration model, higher degree polinomials might be even more advantageous.

**Answer:** Higher degrees were also tested, but this did not improve the results further. We did not include them in our study. One must keep in mind that decreasing the degree of the bias leads to 3 additional parameters per arc. For the whole period of time this is a large number that can change the results dramatically.

30

**R2C64:** 14-13, 14-14: "The additional parameters per arc give room for improving not only the modelled non-gravitational accelerations, but also the gravity field parameters.": I think the connection between "improved" (i.e. more realistic) non-gravitational accelerations and a better quality of the gravity field solution is not only trivial but expected. Anything else is difficult to explain, so the last phrase is possibly unnecessary.

35  **Answer:** We decided to leave the phrase "but also the gravity field parameters" in the paper, because we think it is important

to emphasize (as explained in R2C55).

**R2C65:** 14-14: "works better than a linear function": (1) Please try to use stronger statements. "Works better" can mean numerous things. (2) Do the authors think that reducing the maximum degree of the monthly estimation would allow the linear calibration model to produce results more in-line with GRACE?

**Answer:** (1) We reformulated this: "has a smaller RMSE with respect to GRACE". (2) We do not think that reducing the maximum degree of the monthly estimation would allow the linear calibration model to produce results more in-line with GRACE. We tested this and found out that even though higher degrees (e.g. up to 40) do not contain reliable signal, but they absorb errors in the lower degrees (see R2C29).

**R2C66:** 14-15: "For both, (a) and (b), we also introduced the bias as a constant value or a polynomial of degree 4": Table 4 lists additional scenarios than these two. Were the remaining scenarios analyzed as well? Possibly, this question would not come up if the authors stated explicitly that the previous sentences focused on the per-arc scenarios.

**Answer:** The reviewer is right, we forgot to discuss the scenario with no additional bias or scaling factor, which is now covered. The remaining scenarios relating to bias and scaling factors from Table 4 should be covered:

- scaling factor: 14-10

- bias: constant per arc (pA0) and constant+trend per arc (pA1): this comparison is seperately discussed for the CTAS solutions (14-11) and the monthly solutions (14-14)

- bias: constant global (global0) and polynomial of degree 4 global (global4): this comparison is discussed in 14-15 and 14-6.

For better readability we inserted some line breaks.

**R2C67:** 14-19 to 21: replace "Figure 9 (a) compares ocean mass change derived from the individual monthly solutions to a combination and GRACE, while Fig. 9 (b) shows the solutions estimated with a trend, an annual and a semiannual signal." with "Figure 9 compares ocean mass change derived from the individual solutions, from the combined solution and from GRACE, for (a) the monthly solutions and (b) the solutions estimated with a trend, an annual and a semiannual signal."

**Answer:** Done. We additionally found a mistake in mixing up (a) and (b), which is now corrected.

**R2C68:** 15-Figure 8: It is difficult to read these legends. I suggest the authors find an alternative way to label the plotted lines.

**Answer:** The reviewer is right. Both legends showed the same content, so that we decided to show only one large legend. We now wrote the full words "bias" and "scale" and used the same abbreviations as in Figure 10 and 11. The abbreviations are explained in Table 4.

**R2C69:** 15-1: replace "9 (a)" with "the trend+annual+semiannual" (or a suitable acronym)

**Answer:** Done.

**R2C70:** 15-2: "different receiver settings": Can the authors determine which receiver settings apparently produce higher quality gravity field models?

**Answer:** This would require additional computations that we feel we cannot provide in the framework of time allocated for the revision. It would be very interesting, though. A paper on the "Impact of tracking loop settings of the Swarm GPS receiver on gravity field recovery" has been published by Dahle et al. (2017).

Dahle, C., Arnold, D., and Jäggi, A.: Impact of tracking loop settings of the Swarm GPS receiver on gravity field recovery, Advances in Space Research, https://doi.org/10.1016/j.asr.2017.03.003, 2017.

**R2C71:** 15-5: "...validate our TVG results in other regions": replace "other" with "land"

**Answer:** Done.

**R2C72:** 15-6: "compare": Tense discrepancy: I suggest you use the same verb tense consistently in the same paragraph.

**Answer:** Done.

**R2C73:** 15-7: "the ratio of the root mean square error (RMSE) to the root mean square (RMS) of the GRACE time series": Please motivate this choice for presenting the results and why it was not used before in this article. Edit: this is said a couple of sentences later, but it reads better if the motivation is given before-hand.

**Answer:** The idea was to be able to compare the quality of the solutions of the different areas. This would not be possible with only the RMSE, because the signals have diffferent amplitudes. We reformulated: "We decided to not only compute the RMSE, but tocompute the ratio of the variance (VAR) of the GRACE time series to the RMSE. In this way, we can also compare the quality of the solutions in the different areas.". RMS is replaced with VAR, as suggested in R1C15.

**R2C74:** 16-1: "inverse signal to noise ratio": Why not use the inverse of this measure, which is related directly to the SNR? Noise-to-Signal Ratio is a rather strange metric.

**Answer:** At the beginning of our study we chose 1/SNR, because most of our results had 1/SNR > 1. By this choice the colors could be better distinguished. As our results have improved during the course of the study, we have many results with 1/SNR < 1 (or SNR >1). Hence, the reviewer's advice is very welcome. We changed from 1/SNR to SNR. We chose the new colorbar such as the color green represents 1.

**R2C75:** 16-3: "Figure 10 shows the best 100 solutions": Please explain how the ranking (in which the solutions in figure 10 are sorted) was computed.

**Answer:** We added: "considering SNR for the ocean"

**R2C76:** 16-7: "Greenland and Ganges mass estimates have RMSE/RMS < 1 for trend+annual+semiannual solutions": There are solutions for Gr and Ga in Figure 10 that orange-red, so this statement is not true.

**Answer:** We changed this to "For Greenland and Ganges mass estimates there exist some CTAS solutions with VAR/RMSE > 1".

**R2C77:** 16-8: "they also perform better than Mississippi and Yangtze, when we look at the monthly solutions": This is difficult for the reader to evaluate (if this was determined numerically, please state so). More relevant is the need to provide an explanation for this.

**Answer:** We reformulated this: "and in general, the time series of these two basins have a higher VAR/RMSE than those for Mississippi and Yangtze basins, both for the CTAS and the monthly solutions". This can be seen when comparing the values (colors) of VAR/RMSE (per solution and basin). The explanation is provided as explained in R2C102.

**R2C78:** 16-8: "It is obvious, that modelling non-gravitational accelerations provides better results than not modelling them.": This is not obvious at all. The solutions with ranking 13 to 15 seem to be better for all regions except the Yangtze river basin. The authors should avoid making such over-arching (incorrect) statements, because it gives the sense that they are avoiding a detailed discussion. Therefore, if such large amount of results is presented, the authors need to be prepared to discuss (at least) the outstanding details. This may be a nearly impossible task, regarding some issues, but they still need to be addressed somehow.

**Answer:** We reformulated this part: "Considering monthly solutions, modelling non-gravitational accelerations provides better results than not modelling them. This can be seen in Fig. 11, where only very few solutions with no modelled non-gravitational accelerations are present. The best CTAS solutions for the ocean also have modelled non-gravitational accelerations, whereas for solutions 13 to 15 only empirical accelerations were co-estimated. These have been obtained with a higher VAR/RMSE for the Amazon, Mississippi, Greenland and Ganges basins."

**R2C79:** 16-9, 16-10: "seems to be mandatory": The word "seem" is in conflict with the word "mandatory". Either (e.g.) "seems to be beneficial" or "is mandatory".

**Answer:** The reviewer is right. We chose "is mandatory" because it can be seen in Fig. 10 and Fig. 11 the estimation of a bias is needed to get the best results.

**R2C80:** 16-10: "In general, the results confirm what has been evaluated in Sect. 4.2 to 4.5": As is the case in Sections 4.2 to 4.4, this section lacks interpretation of the results. It is not sufficient to describe the results, they also need to be interpreted.

**Answer:** We agree that these parts read quite technical. We added a few interpretations as suggested in the reviewer's other comments. This paper is a technical one and we want to highlight our methods in particular. As already mentioned, we intend

to publish a paper which focuses more on the physical applications of our results later.

**R2C81:** 16-16: "In Fig. 12 (a) we assumed each GRACE solution to be missing at one time": Please state the month you considered missing and properly motivate this choice. I would guess that removing a month at the extremities, or near an already missing month produces (widely?) different results than cherry-picking. Also in the legend of figure 12.

**Answer:** See R2C90: We changed this to " we assumed each individual monthly GRACE solution to be missing at one time". We did not pick one month to be missing. Every month was assumed to be missing one after another.

**R2C82:** 16-18: "their difference to the real GRACE solution is very close": Please interpret this result.

**Answer:** We wrote: "... we find that they are both very close to the real GRACE solution, which offers two possibilities for bridging monthly gaps in the GRACE time series."

**R2C83:** 16-19: "appears still slightly better": Please avoid using weak statements such as "appears", "slightly" and "better". Such terms may be used but (at least) some sort of quantification is needed to give them context.

**Answer:** We reformulated this sentence: "For most months, the interpolated GRACE time series is closer to the real GRACE solution, which means that it is more reliable to close monthly gaps by interpolating than by using the Swarm solutions."

**R2C84:** 16-19: "close monthly gaps by interpolating than relying on the Swarm solution': Does this means that Swarm is useless in closing small gaps? If so, this is an important result that needs to be announced explicitly.

**Answer:** See R2C83.

**R2C85:** 16-21: "In case of a longer gap between GRACE and GRACE-FO, ocean mass estimates from Swarm will become more important." The new paragraph means the topic under discussion has changed, so it is always a good idea to write things explicitly.

**Answer:** We reformulated this: "will become more important than considering missing monthly solutions".

**R2C86:** 16-23: "mean, trend, annual and semiannual": I suppose a constant term is also estimated.

**Answer:** The reviewer is right. This was meant by "mean". To be more consistent and following R2C38, we replaced this with CTAS.

**R2C87:** 16-25: "Over three years, this would also lead to a degradation of the trend estimate (GRACE and Swarm: 3.5 $\mathrm{mm}$ $\mathrm{yr}^{-1}$ and interpolated GRACE: 4.0 $\mathrm{mm}$ $\mathrm{yr}^{-1}$ ).": Please interpret this result.

**Answer:** We added: "... indicating that Swarm is useful to bridge longer gaps, which will be investigated in the following.".

**R2C88:** 16-30: "35 possibilities": Please state how are the aggregate statistics of all these possibilities calculated (since only one statistic is shown).

**Answer:** Maybe this is a misunderstanding. We hope that this is clear from R2C81 and R2C90, because we do not understand this comment of the reviewer.

**R2C89:** 16-31 to 19-2: "It is obviously better to use our trend+annual+semiannual solution to fill gaps instead of using monthly solutions. However, for a gap of e.g. three months, we get a mean RMSE of 1.1 mm for interpolating existing GRACE solutions compared to 1.5 mm for the trend+annual+semiannual solution": Please interpret these results.

**Answer:** 16-31 is already the interpretation of the table. After 19-2, we added: "... which indicates that in most cases of a three months gap, interpolating the remaining GRACE solutions is closer to GRACE than using the Swarm solutions"

**R2C90:** 19-Figure 12: My understanding is that the blue curve is the result of a constant+trend+annual+semiannual regression, so it is not clear to my why it is not a smooth curve (with continuous first derivative).

**Answer:** The Figure is correct as it is: We assumed each individual monthly GRACE solution to be missing successively. This means we first assume the first month to be missing and use all other data to estimate a constant+trend+annual+semiannual regression. The result is the first blue point. For the second blue point, we assume the second month to be missing and so on. We explained this in more detail now: "In Fig. 13 (a) we assumed each individual monthly GRACE solution to be missing at one time. We then estimated a harmonic time series consisting of CTAS terms from all solutions except for the one that is considered to be missing. After having carried out the regression for each month, this leads to the blue curve."

**R2C91:** 19-Table 9: Please provide an interpretation for the results shown in parenthesis. One would expect that there was a gradual transition from one type of solutions to another, as the number of gap months increased. Looking at the values in parenthesis, this is not the case, and it is difficult to derive any understanding from these results.

**Answer:** It is gradual in terms of RMSE in mm. Percentage is with respect to GRACE interpolated solutions and this does not have to be strictly gradual. Also, the number of investigated cases decreases from left to right. We added "The number of investigated solutions decreases from left to right, as the time span becomes longer." to the caption of Table 9.

**R2C92:** 19-Table 9, caption: (1) blank space missing (2) With "solutions", I assume the authors lumped all (e.g.) 35 possibilities' solutions together and counted those with lower RMSE than interpolated GRACE (relative to Swarm CTAS). I suggest that a longer explanation in the text may avoid these questions.

**Answer:** (1) Corrected. (2) We added: "To derive the value in brackets, we counted the number of CTAS solutions with a lower RMSE than GRACE (interpolated) and computed the relation to the absolute number of CTAS solutions."

**R2C93:** 19-4: This study stands out from the rest of the article quite dramatically and it is severely disconnected from those results. I suggest this section to be removed, unless the authors find a way to motivate it and how it fits with the rest of the

article.

**Answer:** See R2C3.

**R2C94:** 20-11: "As expected prior to launch": Please include the respective reference.

**Answer:** We removed this statement, as simulation studies prior to launch actually expected the signal to be only recovered up to degree 5 to 10 (Wang et al, 2012). We wrote "The degree variances for monthly solutions suggest that the TVG fields are only reliable up to about degree 10-12.", because Figure 3 suggests d/o 10, while, depending on the month, the d/o can be higher.

**R2C95:** 20-14: "for the whole period of time": "under study" (please state explicitly the period start/stop)

**Answer:** Done.

**R2C96:** 20-14: "improves": "improves the agreement with GRACE regarding" (or please find another way to refer to a relative improvement, not an absolute one, which is unknown)

**Answer:** We reformulated this: "We find that this significantly improves the agreement with GRACE regarding ocean mass trend estimates..."

**R2C97:** 20-15 to 20-17: "We investigated different parameterizations and found that an arc length of 30 minutes provides the best results for monthly solutions, while 45 minutes is the best option for the trend+annual+semiannual solutions" Please provide an interpretation for this results.

**Answer:** We are unsure of the reasons for the performarnce of the different arc lengths. The number of observations and unknowns could definitely be important. As stated in R2C1, we plan to write a paper on Swarm ocean mass and its physical interpretation.

**R2C98:** 20-17: "accelerometer bias": Is there a reason for the quotation marks?

**Answer:** See R2C31.

**R2C99: 20-17, 20-18**: "A constant bias per arc and axis works best for monthly solutions and an additional trend parameter is needed for the trend+annual+semiannual approach" (1) Please try to use stronger statements. "Works best" can mean numerous things. (2) Please provide an interpretation for this results.

**Answer:** (1) we reformulated this: "leads to the lowest RMSE with respect to GRACE". (2) Same comment as in R2C97: We are unsure of the reasons for the performarnce of the different arc lengths. The number of observations and unknowns could definitely be important. As stated in R2C1, we plan to write a paper on Swarm ocean mass and its physical interpretation.

**R2C100:** 21-1: remove "by again"

**Answer:** Done.

**R2C101:** 21-4: "appears": Please avoid weak statements, either it is or it is not.

**Answer:** Replaced "appears" with "is".

**R2C102:** 21-4: "signal is too weak": Please provide evidence of this, possibly quoting other studies.

**Answer:** We decided to move the two justifications (R2C102 and R2C103) to Sect. 4.7. The two reasons were given as general possibilities for a weak performance of Swarm. We decided to reformulate the sentence:"In general, the quality of time series of EWH derived from kinematic orbits of Swarm will be affected by (1) The basin size (see Fig. 1) and (2) the signal strength (see Fig. 12).". Fig. 12 in the new document shows the EWH signal for all regions derived from GRACE.

**R2C103:** 21-5: "the basin might be too small": The Ganges basin is smaller than the Mississippi basin, so I don't think this is a good explanation.

**Answer:** See R2C102.

**R2C104:** 21-6, 21-7: The study referred in the preceding sentence has nothing to do with the study referred in the rest of the paragraph. If the authors decide to keep it in the article, please dedicated one paragraph to it (possibly after this paragraph, to be in agreement with the previous section).

**Answer:** The reviewer is right. We moved the sentence about La Niña further to the end of this section.

**R2C105:** 21-7: "In simulation studies": I suggest spending a few more words on this. It was a simulation study in what regards the missing monthly solutions, not in what regards the data used to estimate the gravity field solutions. A less careful reader may easily make this confusion.

**Answer:** We removed "In simulation studies".

**R2C106:** 21-12 to 21-14: How to interpret the results for 6 and 12 months?

**Answer:** The interpretation is that even for 6 and 12 months, GRACE interpolated would be better than actual solutions from Swarm. This -somewhat disappointing- result is related to the fact that during December 2013 and December 2016, ocean mass seems to have evolved rather regularly without interannual changes. As can be seen from Fig. 13 (Fig. 14 in the new manuscript) (based on Wenzel and Schröter), this is an unusual situation and can by no means be assumed to continue in the future.

**R2C107:** 21-16: "(Didova et al., 2016)": The parenthesis on the year only.

**Answer:** Done.

**R2C108:** 21-17: "(Rietbroek et al, 2016)": The parenthesis on the year only.

5 **Answer:** Done.

---

## Author Comment (AC3) · 5 Feb 2018

**Review Answers**

Christina Lück[1], Jürgen Kusche[1], Roelof Rietbroek[1], and Anno Löcher[1]

[1]Institute of Geodesy and Geoinformation, University of Bonn, Bonn, Germany

Dear Editors, dear Reviewer,

we are very grateful for your detailed corrections and annotations. We are confident that your suggestions helped to considerably improve our manuscript. We tried to respond to all comments in the best possible way. The comments are sorted by page-line, which refer to the original, uncorrected document.

**Reviewer 3**

**General Comments:**

**R3C1:** 1-3: replace "estimates" with "monthly snapshots"
**Answer:** Done.

**R3C2:** 1-7: replace ";" with ","
**Answer:** Done.

**R3C3:** 1-9: replace "GRACE missions" with "GRACE and the GRACE-FO mission"
**Answer:** Done.

**R3C4:** 1-10: maybe add "at significantly lower spatial resolution" behind "with respect to GRACE"
**Answer:** The reviewer is right. We should mention the lower resolution of Swarm in the abstract. We decided to mention it in 1-10 at "...substitute missing monthly solutions with Swarm results of significantly lower resolution".

**R3C5:** 1-25: replace "GRACE missions" with "GRACE and the GRACE-FO mission"
**Answer:** Done.

**R3C6:** 2-3: At this point a general reference of the Swarm mission should be given.

**Answer:** Done. Added "(Friis-Christensen et al., 2008)"

**R3C7:** 2-10: Accelerometers provide more information than just air drag.

**Answer:** Changed "... for deriving the drag force" to "... for deriving the non-gravitational accelerations".

**R3C8:** 2-15, 2-16: This is Swarm B, not Swarm C

**Answer:** Done.

**R3C9:** 2-18: The abbreviation GRACE should be introduced at the first occurence.

**Answer:** The reviewer is right. It was already introduced in the abstract.

**R3C10:** 2-25: add "although Jäggi et al. (2009) showed with real GRACE GPS-derived baselines that the benefit will probably be small."

Jäggi, A., G. Beutler, L. Prange, R. Dach, L. Mervart (2009): Assessment of GPS-only observables for Gravity Field Recovery from GRACE. In Observing our Changing Earth, edited by M. Sideris, 133, pp. 113-123.

**Answer:** Done (we added a new sentence: However, Jägge et al. (2009)...).

**R3C11:** 2-27: The right expression for AIUB is "Astronomical Institute of the University of Bern".

**Answer:** Done.

**R3C12:** 2-33: add "gravity" in "monthly Swarm *gravity* solutions".

**Answer:** Done.

**R3C13:** 2-34: add "pressure" in "drag, solar radiation *pressure* and Earth radiation pressure".

**Answer:** Done.

**R3C14:** 3-Table 1: "van den IJssel", not "van Den IJssel"

**Answer:** Done.

**R3C15:** 3-1: Replace "... Swarm satellites which is found to be important to improve the gravity results." with "... Swarm satellites. This has been found to be important to improve the gravity field results".

**Answer:** Done.

**R3C16:** 3-11: It is correct that the quaternions are referring to the mentioned transformation. But I guess they are eventually needed in the processing to relate the satellite rerference frame to the inertial frame?.

**Answer:** We added "During the processing, the satellite reference frame needs to be referred to the inertial frame; this is achieved by multiplying the rotation matrix derived from the star camera data with the Earth rotation matrix (Petit and Luzum,

5  2010)".

**R3C17:** 3-15: If allowed by ESA, the explicit provision of the Swarm macro-model information in the form of a table would be most valuable for the reader and should be given in the manuscript. But I fully understand that ESA has to give permission for this..

10 **Answer:** As the macro model is pretty long and it was provided by ESA for our research, we would prefer if interested persons would directly contact ESA (Christian Siemes).

**R3C18:** 3-18: GOCO05c is complete up to 720.

**Answer:** The reviewer is right. We wanted to say, that we used the model only until d/o 360. We changed the sentence to "...

15 we used the GOCO05c model (Pail et al., 2016) up to degree 360 as a mean background field"..

**R3C19:** 3-20: "except for the atmospheric tides which were chosen such as to be aligned with ITSG Graz solutions": Maybe state explicitly what the difference wrt the GRACE RL05 is.

**Answer:** We believe that the references that are now included should be sufficient

20 (Dahle, C., Flechtner, F., Gruber, C., König, D., König, R., Michalak, G., and Neumayer, K.-H.: GFZ GRACE Level-2 Processing Standards Document for Level-2 Product Release 0005, Tech. rep., Deutsches GeoForschungsZentrum, https://doi.org/10.2312/GFZ.b1(
12020, 2012.

van Dam, T. and Ray, R.: S1 and S2 Atmospheric Tide Loading Effects for Geodetic Applications. Data set/model accessed 2018-01-18, http://geophy.uni.lu/ggfc-atmosphere/tide-loading-calculator.html, Updated October 2010.).

25
**R3C20:** 4-Table 2: add references to products

**Answer:** Done.

**R3C21:** 4-Table 2: EOT11a: Specify up to which degree and order this model is used?

30 **Answer:** We used data from https://doi.pangaea.de/10.1594/PANGAEA.834232. These are gridded $0.125° \times 0.125°$ data with a real and an imaginary part for each tidal constant in units of cm. We converted these data to spherical harmonic coefficients for the main astronomical tides M2, S2, N2, K2, 2N2, O1, K1, P1 and Q1, the long-period tides Mm, Mf, Om1, Om2, Sa, Ssa, Mtm and Msqm, as well as the non-linear constituent M4. The radiational tide S1 is not included here, because it is already considered in the AOD1B RL05 dealiasing product. The processing is consistent for GRACE and Swarm.

35

**R3C22:** 4-5 to 4-7: "In Vielberg et al. (subm.) we compare NRLMSISE-00 to GRACE-derived thermospheric density and derive an empirical correction for this model; this has not yet been applied here.": The statement does not seem to be relevant for this publication. It should be removed.

**Answer:** Vielberg et al. is submitted to the same special issue and we feel a misunderstanding may be possible. Therefore, we prefer to clarify the use of the model here.

**R3C23:** 4-11: add "ITG-" to "...results will be compared to the *ITG*-GRACE solutions.".

**Answer:** Added "ITSG-".

**R3C24:** 4-15: add "ITG-" to "...results are compared to the *ITG*-GRACE solutions.".

**Answer:** Added "ITSG-".

**R3C25:** 5-3: replace "have stronger problems" with "are affected by serious issues".

**Answer:** Done.

**R3C26:** 5-4: Same remark as above (R3C22:"The statement does not seem to be relevant for this publication. It should be removed"): "In the light of recent improvements of empirical thermosphere models (Vielberg et al., subm.)": The statement should rather be eliminated since the original thermosphere models are used in this study.

**Answer:** See R3C22.

**R3C27:** 5-Equation 1: Maybe this formula can just be cited in words. It is quite obvious.

**Answer:** We decided to keep this formula in the paper, because it gives a brief overview and is related to Eq. 2, 3 and 4.

**R3C28:** 5-Equation 2: What about lift forces? Are they included? Is a horizontal wind model used? If yes, which model is used?

**Answer:** Lift forces are included. We added an equation to clarify this. A horizontal wind model is not yet implemented, but this is planned for further research.

**R3C29:** 5-14: What is Aref in this context. Isn't it the surface of one plate of the macro model and the total effect is the sum over all individual plates?

**Answer:** Cited from Doornbos (2011): "Note that the value for the reference area $A_{ref}$ that appears in this equation and similar ones below, should be an agreed value for the entire space vehicle. Its value is not related to the dimensions or orientation of the single panel. It appears in these equations just to make the force coefficient dimensionless." We chose $A_{ref}$ to be the area of the front panel. The value does not matter, because as the reviewer points out, the drag coefficient is computed for each plate. In this computation, the factor $A_i/A_{ref}$ occurs, so that $A_{ref}$ cancels out in the end and the area of each panel is considered with

$A_i$. We have clarified this in the text with "$A_{ref}$ is a reference area that cancels out in the computation of $C_d$ (more precisely in the computation of $C_{D,i,j}$ and $C_{L,i,j}$ , which will be introduced later), where the ratio of the area of each plate to $A_{ref}$ is taken into account.". We decided to refer to Sentman (1961) in the paper instead of Doornbos (2011), because this is the original source.

**R3C30:** 5-15: add "relative to the atmosphere"
**Answer:** Done.

**R3C31:** 5-16: Give the most important keywords how the drag coefficient is computes. Especially how the underlying acco-
10 modation coefficient is chosen should be explicitly mentioned.
**Answer:** We have included an equation for $C_d$ as well as more detailed descriptions.

**R3C32:** 5-19: replace "walls" with "surface".
**Answer:** Done.

15

**R3C33:** 5-12: "Equation 3 accumulates SRP for each of the N plates ...": A similar expression/statement is expected in the previous paragraph on atmospheric drag.
**Answer:** See R3C31.

20 **R3C34:** 6-3, 6-4: "Different from the conventional implementation (Knocke et al., 1988), we expanded these data into a low-degree spherical harmonic representation to account for longitudinal variations" Can the differences to the conventional implementation be quantified? Is it relevant?
**Answer:** Representing the CERES data with low-degree spherical harmonics accounts for longitudinal variations, which is not possible with the approach from Knocke. This can be seen in the Figure below, which is taken from Vielberg et al. (subm.). It
25 shows the ERP acceleration in the along-track direction of the instrument-fixed frame of GRACE A on November 1st 2008. The approach with spherical harmonics (green) is able to represent more variations than the approach from Knocke (red). This is most important in equatorial regions (minima and maxima in the plot). .

**R3C35:** 6-7: Kinematic positions are not original observations but derived from the original GPS observations in a kinematic
30 point positioning, which also provides covariance information of the estimated positions. Has this information been taken into account for the gravity field computation (at least epcoh-wise)?
**Answer:** The reviewer is right, this sentence was misleading. We were talking about positions as "observations" in a least-squares adjustment. We deleted the word "observed". We did not use any covariance information (neither temporal, nor 3x3 fot the estimated positions) as this information is not provided for the official ESA orbits. By now, we do have the 3x3 infor-
35 mation from personal contact with van den IJssel and plan to include it for further studies. Furthermore, our colleagues from

[Figure]

**Figure 1.** ERP acceleration from Knocke (red) and ERP acceleration from spherical harmonic approach (green).

IfE Hanover will provide us with kinematic orbits, including 3x3 covariance information.

**R3C36:** 6-Equation 7: The short-arc approach allows in principle for a very simple quality analysis by comparing the overlap of the positions at the arc boundaries. Since I have never seen such an assessment for short-arc solutions, I am wondering whether it would be worth to be included? A statistics for the different solutions (mean, standard deviation) in the radial, along-track, and cross-track directions would be sufficient.
**Answer:** It is an interesting idea. The reviewer is right in the sense that the fitted boundary positions provide a POD solution simultaneously to the gravity recovery, whose quality is assessed by overlap analysis. However, we refrain from this test here due to three reasons: (1) The paper is on gravity, predominately, not on POD. It would be difficult to say what a good quality threshold would be. (2) Arcs are shorter than in common POD. It would be difficult to compare results. (3) Since the boundary values are reduced from the normal equation systems, it is not trivial to retrieve them without rebuilding and recomputing all of the solutions again.

**R3C37:** 6-16: "y contains all arc-related parameters, which can be eliminated during the estimation.": I guess this means pre-eliminated, i.e. removed from the normal equation system but keeping the impact on the global parameters? Please confirm.
**Answer:** The reviewer is right. We added "... which can be eliminated from the normal equation system" to clarify.

**R3C38:** 6-16: replace "initial and final" with "start and end".
**Answer:** Done.

**R3C39:** 6-21: replace "additionally estimated" with "parametrized"
**Answer:** We reformulated this sentence according to R2C30: "As we aim at a long and stable time series, we additionally parameterize a set of trends and (semi)annual harmonic amplitudes to the constant part for each Stokes coefficient in a single adjustment".

**R3C40:** 7-7: In the case of the short-arc approach, the force modelling errors are to a large extent prevented by the large number of new boundary values estimated at the beginning of each new arc. The allowed jumps between the short arcs act like empirical parameters that may absorb force model errors to a large extent.

**Answer:** The reviewer is right. However, we show in our study that the gravity field is improved when we introduce additional parameters that absorb errors in the modelled non-gravitational accelerations. This paragraph has been rewritten as proposed by Reviewer 2.

**R3C41:** 7-7: "sampling problems": what does this mean?

**Answer:** This sentence has been reformulated: "Due to the presence of errors, e.g. caused by uncertainties in the density model or errors in the macro model, ...".

**R3C42:** 7-8: "it is common to introduce additional parameters": Depending on the chosen arc-length, the parametrization introduces more empirical parameters than other approaches. Is the "accelerometer bias" really needed if accelerometer data are not used?

**Answer:** The reviewer is right. The number of the empirical parameters depends on the chosen arc length. Figure 8 shows the influence of additional bias and scale factors on the ocean mass time series. For most of our best monthly solutions (considering the ocean mass time series) an additional bias has always been co-estimated (as can be seen in Figure 11).

**R3C43:** 7-11: "in that case a polynomial of slightly higher degree could be appropriate": Is this statement based on experiments actually performed? Otherwise it can probably be omitted.

**Answer:** The reviewer is right. We decided to omit this statement. Intuitively, it would make more sense to increase the degree of the polynomial, because we want to allow for changes over the whole time period (compared to additional parameters per arc). Our tests (Figure 8, 10 and 12) do however not show, that a global polynomial of degree 4 is significantly better than a constant global parameter.

**R3C44:** 7-14: replace "controls" with "affects"

**Answer:** Done.

**R3C45:** 7-26: replace "intention" with "application"

**Answer:** Done.

**R3C46:** 8-Figure 1: Maybe a reference should be given from where the boundaries of the river basins are taken.

**Answer:** We added the reference in the caption of Fig.1: "The boundaries are taken from the Food and Agriculture Organization of the United Nations (FAO)".

**R3C47:** 8-Table 3: "estimated until 40": How was this threshold selected. The figure 10 in Dahle et al. (2017) shows that Swarm monthly solutions pass the signal curve in difference degree amplitude plots at about degree 60-70. How would the results presented in this paper change if Swarm would have the chance to contribute more?

**Answer:** The reviewer is right in stating, that the time variable signal from Swarm passes the static field curve at degree 60-70 when looking at the degree amplitudes. We do however believe, that this does not mean, that the Swarm time-variable signal is reliable until d/o 60-70. We think that one should rather look at the crossing point of error (GRACE-Swarm) and signal (GRACE). This is what we did in Figure 3 (dotted gray vs. red). This shows that the Swarm signal is only reliable for low degrees (in this particular plot even only until d/o 10, but in general until about 10-14). For this reason, we chose to evaluate our monthly gravity fields until d/o 10/12/14 (see Table 4). We estimated our gravity fields until higher d/o (20 and 40) because we believe that even though the signal is not reliable in the higher degrees, it might absorb errors in the lower degrees.

**R3C48:** Table 3: "trend per arc (pA1)": Why is this needed? With the large number of boundary conditions in the short-arc approach (and the explicit modeling of non-gravitational accelerations) I would have expected that a bias should be sufficient.

**Answer:** As can be seen in Figure 10, our best trend/annual/semiannual solutions were computed with a bias consisting of a constant value and a trend per arc. However, only a constant value is leads to results that are almost as good. One possible explanation for the additional trend parameter might be the relation between the large number of observations (data from 37 months) to the relatively small number of parameters.

**R3C49:** 8-2: "Swarm alone": Since a high-quality static gravity field model (GOCO05S) based on GRACE and GOCE data has been used, the presented solutions are not Swarm-only solutions. For completeness this should be mentioned at some point in the manuscript. To address in a strict sense what Swarm alone can do, one would have to replace the static field with a long-term solution stemming from Swarm as done, e.g. by Jäggi et al. (2016).

**Answer:** We added "relative to a reference (here GOCO05c)". The reviewer is right in a strict sense. However, our interest is in the time-variability rather than in the static field and no time-variable GRACE (or GOCE) field has been used.

**R3C50:** 8-14: "In a next step, we substitute all degree 1 coefficients to correct for geocenter motion": What does this mean? Are they estimated while stating that they cannot be estimated? Please clarify.

**Answer:** We did not estimate degree 1 coefficients, however, this was not clear from the discussion paper. To clarify, we replaced "substitute" with "add" and we added the sentence "We estimate the spherical harmonic coefficients from degree 2 onward." in 7-7.

**R3C51:** 8-6: "buffer zone": Specify how large this buffer zone is.

**Answer:** This sentence was confusing. It was meant in a way that we chose an ocean mask excluding (i.e. without) a coastal buffer zone. We restrucered the sentence to: "We employed an ocean mask that includes the Arctic ocean and does not have a

coastal buffer zone."

**R3C52:** 8-6: Is the effect of the reduced dealiasing product somehow restored for the presented ocean mass, or does the results just represent the "residual effect". Maybe a short comment should be given at the end of this paragraph.

**Answer:** The reduced dealiasing product has not been added back, because we wanted to concentrate on our results. However, this does not have any influence on the trends, because the GAD product has a trend of zero for the ocean basin. We added Figure 14 (b) and (c) (in the new manuscript), which show ocean mass from GRACE and Swarm with the GAD product added back. We included the paragraph "So far, ocean mass has been shown without adding back the GAD product from the German Research Centre for Geosciences (Flechtner et al., 2015) to our previous time series, since our focus is on comparing estimates and the GAD product has a trend of zero for the ocean basin. Here, for better interpretation, we show ocean mass from GRACE for 2004 to 2013 (Fig. 14 (b)) and ocean mass from Swarm for 2014 to 2016 (Fig. 14 (c)) with the GAD product added back.".

**R3C53:** 8-8: replace "otherwise" with "differently".
**Answer:** Done.

**R3C54:** 9-Table 4: "static part until 20/40/60": Similar remark as for Table 3. A static solution should have sensitivities up to much higher degrees. What happens if the maximum degree is significantly increased, e.g. up to degree 90?
**Answer:** See R3C47. We found that the quality of our time variable solution with trend, annual and semiannual signal does not very much depend on the static part, but rather on how the time-variable part is parameterized.

**R3C55:** 9-5: "leads to our best solution": Did you also compute a solution with non-gravitational accelerations not modelled but still a constant bias per arc estimated? This would show whether the non-gravitational modeling is indeed needed and would show how much you gain by the explicit modeling. I am asking because the experience from GOCE concerning the use of non-gravitational force modeling, where even highly-precise accelerometer data could be used, was rather limited for monthly or bi-monthly gravity field solutions (apart fromd egree 2). See figure 2 in
Jäggi, A., H. Bock, U. Meyer, G. Beutler, J. van den IJssel; 2014: GOCE: assessment of GPS-only gravity field determination. Journal of Geodesy, vol. 89(1), pp. 33-48
**Answer:** The solution with non-gravitational accelerations not modelled but still a constant bias per arc estimated is shown in Figure 5 (gray line). This is now clarified in the caption: "The only difference in IGG(not mod.) is that non-gravitational accelerations were not modelled, but a constant value per arc was still co-estimated.".

**R3C56:** 9-6: "Table 6": I am wondering whether part of the differences is caused by the different resolutions up to different maximum degrees in the individual monthly solutions. In order to decide whether this is relevant you could check for with monthly solutions from IGG resolved up to e.g. d/o 50 or 60 whether the picture is significantly changing or not. If yes, the different cut-off degrees might be responsible for the different noise behaviours.

**Answer:** We used the same cut-of degree (d/o 12) for all individual monthly solutions. This is now clarified in the caption: "The results are based on the time series of Fig. 2".

**R3C57:** 9-7: "after the GNSS receiver update": The Swarm GPS data are heavily affected by ionosphere disturbances (that were partly reduced after the GNSS receiver updates). As a consequence severe systematic errors are visible in monthly solutions along the geomagnetic equator (depending on ionosphere conditions). This aspect is not mentioned in the paper. Was it found to be not relevant?

**Answer:** We did not investigate this effect in this study. It probably only has a minor effect on our ocean mass tim series, because we look at the average over the whole ocean. This is however an interesting point that we will work on in the future.

**R3C58:** 9-7: The updates of the tracking loop settings and their impact on the gravity field solutions are not mentioned in this manuscript. Are they not relevant for the derivation of ocean mass. See table 1 in Dahle et al. (2017).

Dahle, C., D. Arnold, A. Jäggi (2017): Impact of tracking loop settings of the Swarm GPS receiver on gravity field recovery. Advances in Space Research, 59(12), 2843-2854

**Answer:** We have not yet investigated the impact of tracking loop settings on the derivation of ocean mass. This is certainly foreseen for the future. We referred to the paper that the reviewer mentioned "The impact of tracking loop updates on gravity field recovery is discussed in Dahle et al. (2017)."

**R3C59:** 9-18: "our IGG solution for May 2016": Which IGG solution is shown here? The best solution based on 30 min arc-length with modelled accelerations and a bias estimated per arc in all directions? I guess the solid lines in Fig. 3 (GRACE, IGG) represent differences wrt to a static field. Please explicitly state wrt to which model the differences are hown.

**Answer:** Yes, "IGG solution" always refers to the solution mentioned in Table 3, as mentioned in 8-8: "If not stated otherwise, we used the parameterization in Table 3 for monthly ocean mass...". We included "with respect to our reference field GOCO05c".

**R3C60:** 9-20: maybe add a statement like: "... due to the much lower precision of the GPS data compared to the GRACE inter-satellite K-Band ranging".

**Answer:** Done.

**R3C61:** 9-21: "The formal errors (dotted black line) appear to be too optimistic, as they are always lower than the difference between GRACE and Swarm." Maybe a statement could be given how to address this in future work.

**Answer:** Added "This will be addressed in the future by including realistic covariance information of the kinematic orbits." Our colleagues from IfE Hanover are working on this topic.

**R3C62:** 10-Table 5: For AIUB solutions have been derived based on original and screened kinematic orbits. Which solution has been used to compute the ocean mass time series?

**Answer:** The results are based on the screened kinematic orbits. We have added this information in Table 5.

**R3C63:** 10-Table 5: Is this the max d/o of the solutions provided or the max d/o of the solutions used?

**Answer:** This is the maximum d/o provided, we clarified this in the caption. For the comparisons we used d/o 12. We corrected a mistake: Swarm ASU solutions are available until d/o 40 (not 50).

**R3C64:** 11-Table 7: How was GRACE treated to derive the numbers provided in the table? Cut-off at the same degree than for the Swarm solutions?

**Answer:** We used the same cut-of degree (d/o 12) for all individual monthly solutions. This is now clarified in the caption: "The results are based on the time series of Fig. 2"..

**R3C65:** 11-18: Are the gaps in the GRACE series also introduced in the Swarm series to estimate most comparable trends and amplitudes, or is the amount of data points different for the GRACE and the Swarm series?

**Answer:** We did not consider the monthly gaps for the Swarm trends, amplitudes and phases. This is now also included indicated by values in brackets.

**R3C66:** 12-Figure 4: Is this in the along-track direction of the local orbital frame or in the instrument-fixed frame?

**Answer:** This is the along-track direction of the instrument-fixed frame.

**R3C67:** 12-Figure 4: As mentioned in a previous comment, it would also be interesting to have a solution with non-gravitational accelerations not modelled but a constant bias per arc still estimated? Only one different setting in the processing would then be addressed and it would become clear whether the improvements are really due to the non-gravitational force modeling (or rather due to the estimation of additional empirical parameters per arc).

**Answer:** See R3C55.

**R3C68:** 12-4, 12-5: Please clearly declare whether the modelled accelerations are shown without the estimated scale or bias parameters.

**Answer:** Figure 4 shows the modelled accelerations without any bias or scale factors. We have now clarified this in the caption: "The black curve shows the ACC3CAL_2_ product from Siemes et al. (2016), while the red curve shows our modelled non-gravitational accelerations (without applying any bias or scale factors).". A bias has however been co-stimated in the gravity field estimation process for Figure 5 (IGG (mod.)). We tried to mention this in 8-8: "If not stated otherwise, we used the parameterization in Table 3 for monthly ocean mass or ocean mass from a direct estimation of trend, annual and semiannual

signal terms is shown.", but we now emphasized it as explained in R3C55, R3C59, R3C67, R3C73.

**R3C69:** 12-7: "and it also improves the trend estimate as can be seen in Table 8": How can this be seen? Was there a solution with trend estimation when not using accelerometer data? Maybe I just missed it?

**Answer:** Yes, we talk about the ocean mass trend here. It can be seen in Table 8. The trend from GRACE is 3.5 mm/yr. Swarm with modelled non-gravitational accelerations (parameterization as in Table 4 (monthly)) has a trend of 3.3 mm/yr and the solution without modelled non-gravitational accelerations (still same parameterization, also with a constant bias) has a trend of 4.0 mm/yr.

**R3C70:** 13-Table 8: "same parameterization as IGG": According to Table 4 the parametrization of the bias is different. Please clarify.

**Answer:** No, for both solutions a constant bias per arc was co-estimated, so the caption of Table 8 is correct. I think the reason for the misunderstanding and the comments R3C55, R3C67 and R3C73 is Table 4. It should be read in the following way: we testet arc lengths of 30 minutes, 45 minutes and 60 minutes (column 1). We tested non-gravitational accelerations either modelled or not (column 2),... Table 4 should not be read row-wise. A solution consists of on choice in each column. We tried to clarify this with a new caption: "Parameterizations that have been tested in this study. This table should not be read row-wise. It lists all possible choices for each heading. One solution can consist of any combination of the entries, for example: a monthly solution with an arc length of 60 minutes, modelled non-gravitational accelerations, a constant global bias, no scaling factor, max. d/o estimated: 40, evaluated until d/o 10.".

**R3C71:** 14-6: Could the authors give some explanation why the 30 min case is worst for the solution based on trends, annual and ssemi-annual signals, and essentially the opposite for the monthly solutions?

**Answer:** We are not sure about the reasons. It could be related to the ratio of the number of observations to the number of unknown parameters. Both differ extremely for the two cases.

**R3C72:** 16-18: "have RMSE/RMS<1": Maybe one should say that there exist some solution types with RMSE/RMS < 1. Especially for Ganges there seem to be many solution types with a ratio larger than 1.

**Answer:** This is right, we changed this to "For Greenland and Ganges mass estimates there exist some CTAS solutions with VAR/RMSE > 1 ". We use the variance instead of the RMS as suggested in R1C15.

**R3C73:** 16-18: "modelling non-gravitational accelerations provides better results than not modelling them": As mentioned before, I would like to see that this is not only due to the estimation of additional empirical bias parameters.

**Answer:** See R3C55.

**R3C74:** 16-19: Delete "Nevertheless".

**Answer:** Done.

**R3C75:** 20-8: maybe it should be recapitulated up to which degree the Swarm solutions are solved for in this study.

5  **Answer:** As explained before, we believe that the time-variable part of the Swarm solutions is only reliable until about d/o 10-15. This is we we chose d/o 10, 12 and 14 for investigations. We did estimate our gravity fields until degrees 20,40 (and 60 for trend, annual and semiannual solutions) because higher degrees can absorb errors in the lower degrees.

**R3C76:** 20-12: replace "degree about" with "about degree".

10  **Answer:** Done.